# ☀ SPARK: STEPWISE PROCESS-AWARE REWARDS FOR REFERENCE-FREE REINFORCEMENT LEARNING

## ABSTRACT

Process reward models (PRMs) that provide dense, step-level feedback have shown promise for reinforcement learning, yet their adoption remains limited by the need for expensive step-level annotations or ground truth references. We propose SPARK–a three-stage framework where in the first stage a generator model produces diverse solutions and a verifier model evaluates them using parallel scaling (self-consistency) and sequential scaling (meta-critique). In the second stage, we use these verification outputs as synthetic training data to fine-tune generative process reward models, which subsequently serve as reward signals during training. We show that aggregating multiple independent verifications at the step level produces training data for process reward models that surpass ground-truth outcome supervision—achieving 67.5 F1 on ProcessBench (a benchmark for identifying erroneous steps in mathematical reasoning) compared to 66.4 for reference-guided training and 61.9 for GPT-4o. In the final stage, we apply our generative PRM with chain-of-thought verification (PRM-CoT) as the reward model in RL experiments on mathematical reasoning, and introduce format constraints to prevent reward hacking. Using Qwen2.5-Math-7B, we achieve 47.4% average accuracy across six mathematical reasoning benchmarks, outperforming ground-truth-based RLVR (43.9%). Our work enables reference-free RL training that exceeds ground-truth methods, opening new possibilities for domains lacking verifiable answers or accessible ground truth.

## 1 INTRODUCTION

Large language models (LLMs) have demonstrated impressive capabilities across diverse tasks, from achieving gold-medal performance at the International Mathematical Olympiad to autonomous agentic coding (Castelvecchi, 2025; Luong & Lockhart, 2025; Yang et al., 2024b; Hurst et al., 2024; Anthropic, 2025). Despite these achievements, LLMs still struggle with complex multi-step reasoning and long-horizon problem solving (Kambhampati et al., 2024; Yao et al., 2024; Valmeekam et al., 2024). Recent breakthroughs like OpenAI's o1 and DeepSeek's R1 demonstrate that reinforcement learning (RL) post-training can significantly enhance reasoning capabilities beyond supervised fine-tuning alone (Jaech et al., 2024; Guo et al., 2025), as RL enables models to explore diverse solution paths and learn from feedback rather than imitation (Chu et al., 2025).

While RL post-training shows promise, current approaches rely on verifiers that require ground truth references. Traditional methods rely on either discriminative verifiers that provide binary correctness signals (Cobbe et al., 2021) or rule-based verifiers using exact answer matching (RLVR) (Guo et al., 2025; Hu et al., 2025), both offering only sparse, outcome-level rewards. Recent advances introduce Process Reward Models (PRMs) that provide denser, step-level feedback to improve training stability and credit assignment (Lightman et al., 2023; Wang et al., 2024; Uesato et al., 2022), including co-evolving approaches like TANGO (Zha et al., 2025) and PRIME (Yuan et al., 2024) that jointly train the verifier alongside the policy model. However, these approaches fundamentally depend on ground truth references—TANGO trains its verifier using gold standard solutions, while PRIME requires outcome-level correctness labels to train its PRM (Zha et al., 2025; Yuan et al., 2024). This dependency severely limits RL's applicability to domains where ground truth is unavailable, requires expensive expert annotation, or lacks clear verification criteria, such as creative

Corresponds to: salman@cs.ucla.edu, srgnt@amazon.com

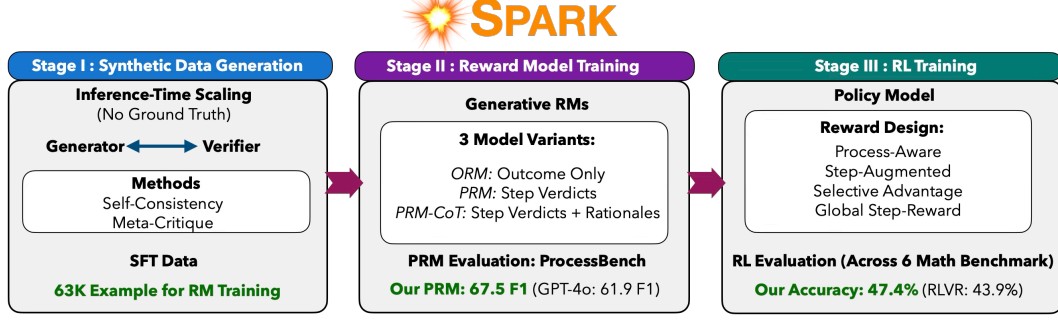

Figure 1: **SPARK**: A three-stage pipeline for reference-free RL training with generative process reward models. Stage I: Generate synthetic verification data using inference-time scaling methods (self-consistency and meta-critique) without ground truth through a multi-scale generator-verifier framework. Stage II: Train three generative reward model variants (ORM, PRM, PRM-CoT) via supervised fine-tuning on the synthetic data. Stage III: Apply trained PRMs in RL with GRPO using different reward designs.

writing, research ideation, long-horizon planning, or complex agentic tasks (Bowman et al., 2022). The challenge becomes: *How can we train effective process reward models that provide dense, step-level feedback without requiring any ground truth references, enabling RL to scale beyond domains with verifiable answers?*

The recent success of inference-time scaling methods offers a promising direction for addressing this challenge. These approaches improve LLM reasoning by allocating additional computation at test time rather than during training (Ke et al., 2025; Snell et al., 2025; Brown et al., 2024). Parallel scaling methods like self-consistency demonstrate that aggregating multiple independent reasoning paths through majority voting significantly improves accuracy over single-path generation (Wang et al., 2023). Sequential scaling methods, such as self-refinement approaches, show that LLMs can iteratively critique and improve their own outputs without external supervision (Madaan et al., 2023; Saunders et al., 2022). These inference-time techniques have proven highly effective, with recent work showing that optimal test-time compute scaling can outperform simply increasing model parameters (Snell et al., 2025). This raises a critical insight: if LLMs can improve reasoning by aggregating multiple solution attempts (self-consistency) or iteratively refining outputs (self-critique) at inference time without ground truth, can we leverage the same capabilities to generate synthetic verification data for training generative process reward models?

In this work, we propose **SPARK**, a reference-free framework that leverages inference-time scaling methods to generate synthetic step-level verification data without any ground truth references (see Figure 1). We employ a multi-scale generator-verifier framework where a generator model produces diverse solution attempts and a verifier model evaluates them using parallel (self-consistency) and sequential (meta-critique) scaling techniques. Our key insight is that aggregating multiple independent verifications at the step level can produce training data that rivals or exceeds ground-truth supervision quality. We demonstrate that PRMs trained using this approach enable stable RL training while systematically identifying and addressing multiple reward exploitation patterns that emerge when using generative PRMs as reward signals—challenges that prior work has not comprehensively explored (Zha et al., 2025; Cui et al., 2025). The contributions of our **SPARK** framework include:

(1) A reference-free framework for generating high-quality step-level verification data using inference-time scaling, eliminating the need for ground truth or human annotation (Section 2).

(2) Comprehensive evaluation on ProcessBench (Zheng et al., 2025), a benchmark for identifying erroneous steps in mathematical reasoning, showing that PRMs trained with our synthetic data achieve 67.5 F1, outperforming those trained with outcome ground-truth access (66.4 F1) and surpassing GPT-4o by 5.6 points (Section 3).

(3) We further demonstrate that our reference-free PRMs enable stable RL training that matches or exceeds ground-truth-based RLVR when properly constrained, while systematically identifying and

addressing reward exploitation patterns unique to generative PRMs—opening new possibilities for RL in domains without ground truth or verifiable answers (Section 4).

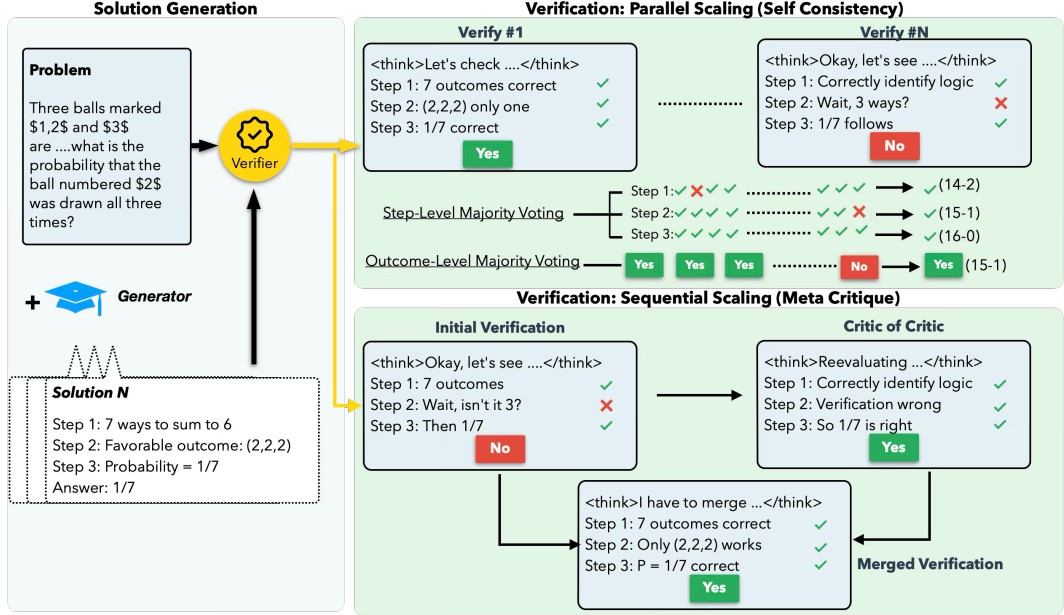

Figure 2: Multi-scale generator-verifier framework for synthetic verification data generation. The generator produces multiple solutions per problem, and the verifier evaluates them without ground truth using different inference-time scaling methods. **Parallel scaling (Self-Consistency):** Generates multiple independent verifications and aggregates them through either outcome-level majority voting (voting on final Yes/No verdicts) or step-level majority voting (voting on each step's correctness). **Sequential scaling (Meta-Critique):** Generates an initial verification, critiques it to identify errors, and merges both into a refined verification.

## 2 GENERATING SYNTHETIC VERIFICATION DATA FOR PRM TRAINING

In this section, we describe how to generate synthetic step-level verification data for training PRMs using our multi-scale generator-verifier framework, which leverages inference-time scaling methods to produce high-quality labels without ground truth. Given a problem $q$ and an LLM-generated solution $s = (s_1, s_2, ..., s_n)$ with $n$ reasoning steps in "Step" tags, we produce verification labels $v^{(j)} = (v_1^{(j)}, v_2^{(j)}, ..., v_n^{(j)})$ where $v_i^{(j)} \in \{\text{correct}, \text{incorrect}\}$ for each step $s_i$ (see prompts in Appendix §F). Our framework employs Qwen-2.5-14B-Instruct (generator) to produce $M = 8$ diverse solutions per problem via temperature sampling, and Qwen-3-32B-Instruct (verifier) to evaluate each solution through the methods illustrated in Figure 2. We leverage two categories of inference-time scaling: parallel scaling through self-consistency and sequential scaling through meta-critique. Detailed implementation and notation are provided in Appendix §A. We implement the following inference-time scaling methods to generate verification data.

**Self-Consistency** generates $N = 16$ independent verifications for each problem-solution pair $(q, s)$ and aggregates them through majority voting. We implement two variants:

*(1) Outcome-level consistency:* Let $y^{(j)} \in \{\text{Yes}, \text{No}\}$ denote the final verdict of verification $v^{(j)}$. We determine the consensus verdict as: $y^* = \arg\max_{y \in \{\text{Yes}, \text{No}\}} \sum_{j=1}^{N} \mathbb{1}[y^{(j)} = y]$. We then randomly select one verification $v^{(k)}$ where $y^{(k)} = y^*$.

*(2) Step-level consistency:* Let $v_i^{(j)} \in \{\text{correct}, \text{incorrect}\}$ denote the judgment of step $i$ in verification $j$. For each step $i$, we determine the consensus judgment: $v_i^* = \arg\max_{v \in \{\text{correct}, \text{incorrect}\}} \sum_{j=1}^{N} \mathbb{1}[v_i^{(j)} = v]$. This produces a consensus verification pattern $(v_1^*, v_2^*, ..., v_n^*)$. We then randomly select one verification $v^{(k)}$ where $v_i^{(k)} = v_i^*$ for all $i \in \{1, ..., n\}$.

**Meta-Critique** sequentially refines a single verification (McAleese et al., 2024; Yang et al., 2025; Saunders et al., 2022; Gou et al., 2024). The verifier performs three steps: (1) generates initial verification $v_{\text{init}}$ for problem-solution pair $(q, s)$; (2) critiques this verification to identify errors—missed mistakes, incorrectly flagged steps, or flawed reasoning—producing $\kappa = \text{Critique}(q, s, v_{\text{init}})$; and (3) merges the critique with the initial verification into $v_{\text{final}} = \text{Merge}(v_{\text{init}}, \kappa)$. All three steps are performed by the same verifier model using different prompts (see Appendix §F). The refined verification $v_{\text{final}}$ serves as our training example.

**Hybrid (Outcome Consistency + Meta-Critique)** combines parallel and sequential scaling: first applies outcome-level consistency to select the best verification from $N = 16$ independent attempts, then applies meta-critique to refine it further, producing our final training data.

Through these methods, we generate training datasets $\mathcal{D}$ containing problem-solution-verification triples that enable training generative PRMs without ground truth, as described in the next section.

# 3 TRAINING GENERATIVE PROCESS REWARD MODELS

## 3.1 REWARD MODEL TRAINING SETUP

**Dataset.** We use Skywork-OR1-RL-Data (He et al., 2025a;b), randomly selecting 8,000 math problems with mixed difficulty levels. Using the multi-scale generator-verifier framework, we generate 8 solution attempts per problem (64K problem-solution pairs total). We then apply each method from Section 2 to generate verification data: single verification produces 1 verification per pair; self-consistency methods (outcome/step-level) generate 16 verifications then aggregate to select one; meta-critique generates and refines 1 verification; hybrid combines outcome consistency's selection with meta-critique's refinement. This process yields 63K verification examples per method after filtering, creating training datasets $\mathcal{D}$ for each reward model variant.

**Generative Reward Models.** Unlike prior work that trains discriminative reward models outputting numerical scores (Cobbe et al., 2021; Lightman et al., 2023; Wang et al., 2024), we follow Zhang et al. (2025a) and train generative reward models using next-token prediction on synthetic verification data from Section 2. We train three variants on dataset $\mathcal{D}$:

1. **ORM** outputs only final verdict $y \in \{\text{Yes}, \text{No}\}$ given $(q, s)$. Training: $\mathcal{D}_{\text{ORM}} = \{(q, s, y)\}$.

2. **PRM** outputs step-by-step judgments $(v_1, ..., v_n, y)$ where $v_i \in \{\text{correct}, \text{incorrect}\}$ for step $i$. Training: $\mathcal{D}_{\text{PRM}} = \{(q, s, (v_1, ..., v_n, y))\}$.

3. **PRM-CoT** outputs verification rationales with judgments $((\tau_1, v_1), ..., (\tau_n, v_n), y)$ where $\tau_i$ explains step $i$'s correctness. Training: $\mathcal{D}_{\text{PRM-CoT}} = \{(q, s, ((\tau_1, v_1), ..., (\tau_n, v_n), y))\}$.

All models are fine-tuned from Qwen2.5-14B-Instruct (Qwen, 2024) for 3 epochs with learning rate $5 \times 10^{-6}$. Detailed specifications in Appendix §B.

**Evaluation Protocol.** Following previous work (Zha et al., 2025; Yang et al., 2025; Khalifa et al., 2025), we evaluate our PRMs on ProcessBench (Zheng et al., 2025), a benchmark for identifying erroneous steps in mathematical reasoning. ProcessBench requires models to identify the earliest incorrect step or conclude all steps are correct. The benchmark contains 3,400 test cases across GSM8K, MATH, OlympiadBench, and Omni-MATH (grade-school to Olympiad difficulty), with solutions from 12 models annotated by human experts. We report F1 scores (harmonic mean of accuracies on correct and incorrect solutions) to balance over-criticism and under-detection of errors.

**Baselines for PRM Evaluation.** We compare PRMs trained with our inference-time scaling methods against: **(1) Single Verification** – verifier generates one verification without scaling (baseline); **(2) Reference-Guided** – verifier has ground truth answer $a^*$ when verifying $(q, s)$, providing additional context for verification (Zhang et al., 2025a; Zheng et al., 2023); **(3) LLM Critics** – GPT-4o (Hurst et al., 2024) and Qwen2.5-72B-Instruct (Qwen, 2024) as off-the-shelf critics following Zheng et al. (2025).

## 3.2 RESULTS: PROCESSBENCH EVALUATION

Figure 3 presents F1 scores on ProcessBench for our two PRM variants trained with data from different inference-time scaling methods.

**Inference-time scaling surpasses reference-guided approach.** Among all inference-time scaling methods, *step-level consistency achieves the highest performance across both PRM variants*. On ProcessBench, step-level consistency achieves F1 scores of 67.5 (PRM) and 65.7 (PRM-CoT), surpassing reference-guided scores of 66.4 and 63.2 respectively. The hybrid approach (Meta-Critique + Outcome Consistency) surpasses reference-guided performance for both PRM (66.9 vs 66.4) and PRM-CoT (63.7 vs 63.2). Additionally, outcome consistency surpasses reference-guided approach for PRM (66.6 vs 66.4) and PRM-CoT (65.0 vs 63.2). This demonstrates that PRMs trained using various inference-time scaling methods outperform those trained with ground truth access.

**All scaling methods improve over single verification.** Every inference-time scaling method substantially improves over the single verification baseline, validating our core hypothesis that inference-time scaling is effective for synthetic verification data generation. Improvements range from +1.3 to +7.0 F1 points, with step-level consistency achieving the highest gains (PRM: $63.9 \rightarrow 67.5$, PRM-CoT: $59.8 \rightarrow 65.7$).

**SPARK-trained PRMs outperform frontier LLM critics.** Our PRMs trained with SPARK significantly outperform both GPT-4o (61.9 F1) and Qwen2.5-72B-Instruct (61.2 F1).

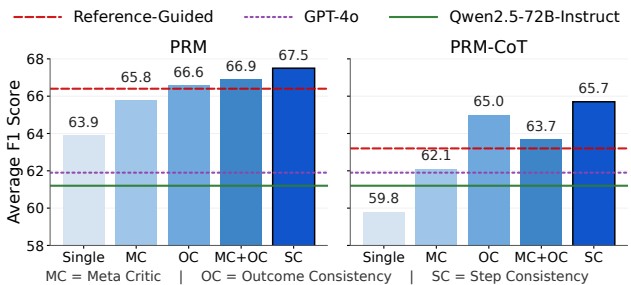

Figure 3: Average F1 scores on ProcessBench for PRM variants trained using synthetic data from different inference-time scaling methods. Leftmost bars show Single Verification baseline (no scaling). All PRMs are fine-tuned from Qwen2.5-14B-Instruct.

Even single verification baseline—without scaling—achieves higher scores for PRM (63.9) and comparable performance for PRM-CoT (59.8). With step-level consistency, the gap widens: PRM reaches 67.5 (+5.6 over GPT-4o) and PRM-CoT reaches 65.7 (+3.8 over GPT-4o). This demonstrates that training specialized 14B PRMs with SPARK is more effective than using general-purpose frontier models as critics.

Our best PRM (step-level consistency) achieves 67.5 F1, outperforming existing open-source PRMs (Qwen2.5-Math-7B-PRM800K at 56.5 F1). Detailed comparisons in Table 2 in Appendix §C.

> **Takeaway:** Step-level consistency—aggregating multiple independent verifications at the step level—enables PRMs to surpass both reference-guided training and frontier critics like GPT-4o. This demonstrates that inference-time scaling provides a viable alternative to ground truth supervision for training high-performance generative process reward models.

## 4 REINFORCEMENT LEARNING WITH PROCESS REWARDS

### 4.1 RL METHODOLOGY

**Policy Optimization with GRPO.** We employ Group Relative Policy Optimization (Shao et al., 2024), generating $M = 16$ solutions per problem and optimizing the following objective:

$$J(\theta) = \mathop{\mathbb{E}}_{q,\{o_i\}_{i=1}^M} \left[ \frac{1}{M} \sum_{i=1}^{M} \frac{1}{|o_i|} \sum_{t=1}^{|o_i|} \min\left( r_t(\theta)\hat{A}_{i,t}, \text{clip}(r_t(\theta), 1-\epsilon, 1+\epsilon)\hat{A}_{i,t} \right) \right] - \beta D_{\text{KL}}(\pi_\theta \| \pi_{\text{ref}}),$$

where $o_i$ is the $i$-th solution, $o_{i,t}$ is the $t$-th token in $o_i$, $r_t(\theta) = \frac{\pi_\theta(o_{i,t}|q,o_{i,<t})}{\pi_{\theta_{\text{old}}}(o_{i,t}|q,o_{i,<t})}$ is the importance ratio, and $\hat{A}_{i,t}$ is the group-normalized advantage for token $t$ in solution $i$.

Table 1: Performance comparison of SPARK-trained PRMs with existing methods on mathematical reasoning benchmarks. Results show pass@1 accuracy (%) with greedy decoding.

| Model | MATH500 | AIME'24 | AIME'25 | AMC'23 | OlympiadBench | MinervaMath | Avg. |
|---|---|---|---|---|---|---|---|
| *Frontier LLMs* | | | | | | | |
| GPT-4o (Hurst et al., 2024) | 76.6 | 9.3 | – | 47.5 | 43.3 | 36.8 | – |
| o1-preview (Jaech et al., 2024) | 85.5 | 44.6 | – | 90.0 | – | – | – |
| o1-mini (Jaech et al., 2024) | 90.0 | 56.7 | – | 95.0 | 65.3 | – | – |
| *Open-source LLMs (Large)* | | | | | | | |
| QwQ-32B-Preview (Team, 2024) | 90.6 | 50.0 | 33.3 | 77.5 | 61.2 | – | – |
| Llama-3.1-70B-Inst (Dubey et al., 2024) | 68.0 | 13.3 | – | 42.5 | 29.4 | 37.1 | – |
| Qwen2.5-Math-72B-Inst (Yang et al., 2024a) | 82.6 | 23.3 | – | 70.0 | 49.0 | – | – |
| *Open-source LLMs (Small)* | | | | | | | |
| Llama-3.1-8B-Inst (Dubey et al., 2024) | 51.9 | 3.3 | 3.3 | 22.5 | 15.1 | – | – |
| Qwen2.5-7B-Inst (Qwen, 2024) | 75.5 | 10.0 | 6.7 | 52.5 | 35.5 | – | – |
| Qwen2.5-Math-7B-Inst (Yang et al., 2024a) | 83.6 | 16.7 | 10.0 | 62.5 | 41.6 | 34.6 | 41.5 |
| *RL with Process Rewards (7B)* | | | | | | | |
| TANGO[†] (Zha et al., 2025) | 82.4 | 26.7 | 23.3 | 70.0 | 45.3 | – | – |
| PRIME[†] (Cui et al., 2025) | 78.2 | 20.0 | 13.3 | 70.0 | 40.3 | 39.3 | 43.6 |
| RLVR | 83.7 | 26.7 | 16.7 | 59.1 | 40.0 | 37.5 | 43.9 |
| *Reference-Free Gen-PRMs (Qwen2.5-Math-7B)* | | | | | | | |
| SFT (Baseline) | 79.0 | 3.3 | 3.3 | 52.5 | 34.7 | 34.2 | 34.5 |
| PRM (Process-Aware) (Ours) | 82.8 | 26.7 | 16.7 | 62.5 | 38.7 | 37.1 | 44.1 |
| **PRM-CoT (Process-Aware) (Ours)** | **85.4** | **30.0** | **20.0** | 66.3 | 42.7 | 40.1 | **47.4** |

[†] Results from corresponding papers. **Bold** values indicate best performance among 7B models. Extended results with Pass@k metrics in Table 3.

**Reward Formulations.** We investigate four reward mechanisms leveraging our generative PRMs:

**(1) Process-Aware Rewards.** Our PRMs evaluate step-by-step correctness (PRM directly, PRM-CoT with verification rationales) before providing final verdict $y \in \{\text{Yes}, \text{No}\}$. We extract this verdict with format validation:

$$r_{\text{process}}(s) = \mathbb{1}[\text{valid format}] \cdot y, \quad \hat{A}_{\text{process}}^{(i)} = \frac{r_{\text{process}}^{(i)} - \mu_G}{\sigma_G},$$

where $y \in \{0, 1\}$ is the verification verdict, $\mathbb{1}[\text{valid format}]$ ensures proper output structure (single `<answer>` tag, single `\boxed{}` expression, no post-answer content), and $\mu_G, \sigma_G$ are group-level statistics. We term this *"process-aware"* because the final verdict implicitly aggregates step-level verification through autoregressive dependency.

**(2) Step-Augmented Process Rewards.** We explicitly incorporate step-level signals by augmenting the process-aware reward with step-average scores. Given solution $s$ with $n$ steps where PRM marks $k$ steps as correct:

$$r_{\text{step-aug}}(s) = \mathbb{1}[\text{valid format}] \cdot \left[ 0.4 \cdot \frac{k}{n} + 0.6 \cdot y \right], \quad \hat{A}_{\text{step-aug}}^{(i)} = \frac{r_{\text{step-aug}}^{(i)} - \mu_G}{\sigma_G},$$

where $\frac{k}{n}$ is the step correctness ratio and $y$ is the process-aware verdict.

**(3) Selective Advantage.** To avoid penalizing correct steps in failed solutions and rewarding incorrect steps in successful solutions, we selectively zero misaligned advantages. For token $t$ in step $j$ with verdict $c_j \in \{\text{correct}, \text{incorrect}\}$:

$$\hat{A}_{\text{selective}}^{(i,t)} = \begin{cases} \hat{A}_{\text{process}}^{(i)} & \text{if } (\hat{A}_{\text{process}}^{(i)} \geq 0 \wedge c_j = \text{correct}) \vee (\hat{A}_{\text{process}}^{(i)} < 0 \wedge c_j = \text{incorrect}) \\ 0 & \text{otherwise.} \end{cases}$$

**(4) Global Step-Reward.** Following Zha et al. (2025), we blend process-aware and step-level advantages. For solution $i$ with $K_i$ steps, normalized step rewards are $r_{\text{step}}^{(i,k)} = c_k / K_i$ where $c_k \in \{+1, -1\}$ for correct/incorrect steps. Each token in step $k$ receives cumulative advantages:

$$\hat{A}_{\text{step}}^{(i,t)} = \sum_{j=k}^{K_i} \frac{r_{\text{step}}^{(i,j)} - \mu_{G,\text{step}}}{\sigma_{G,\text{step}}}, \quad \hat{A}_{\text{global}}^{(i,t)} = 0.8 \cdot \hat{A}_{\text{process}}^{(i)} + 0.2 \cdot \hat{A}_{\text{step}}^{(i,t)}.$$

where $\mu_{G,\text{step}}, \sigma_{G,\text{step}}$ are computed globally across all steps from all $M$ solutions—hence "Global Step-Reward."

**Baseline: RLVR.** For comparison, we include Reinforcement Learning from Verifiable Rewards (RLVR), which uses ground truth for verification:

$$r_{\text{RLVR}}(s) = \mathbb{1}[\text{final answer matches ground truth}], \quad \hat{A}_{\text{RLVR}}^{(i)} = \frac{r_{\text{RLVR}}^{(i)} - \mu_G}{\sigma_G}.$$

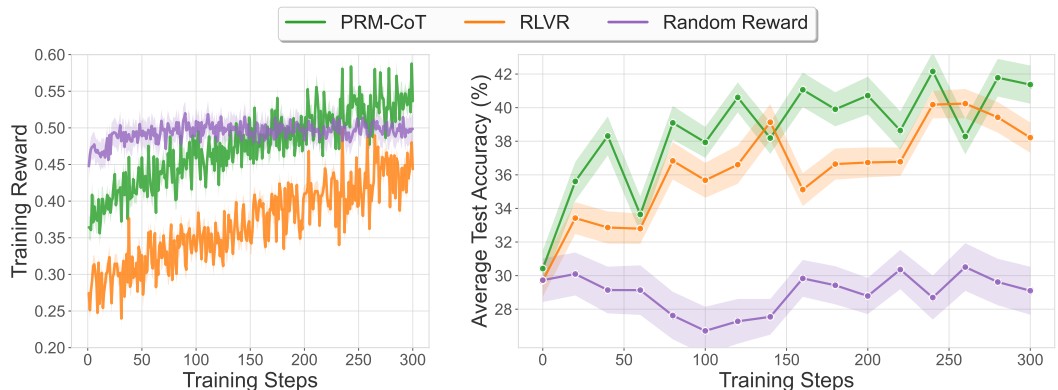

Figure 4: Comparison of reference-free PRM-CoT, ground-truth RLVR, and random rewards. **Left:** Training rewards for (1) PRM-CoT with process-aware rewards (Section 4.1), (2) RLVR with ground-truth answer verification, and (3) random rewards via coin flip (50% probability) independent of correctness. **Right:** Average test accuracy on MATH-500, AIME 2024, and AIME 2025. PRM-CoT consistently outperforms RLVR while spurious random rewards fail to improve from baseline.

## 4.2 RL EXPERIMENTAL DETAILS

We use Qwen2.5-Math-7B (Yang et al., 2024a) as our policy model. Following Zha et al. (2025), we first perform SFT on 113K problems from Eurus-2-SFT-Data (Cui et al., 2025) with structured solutions (step-by-step reasoning in `<step>` tags, answers in `<answer>` tags) generated by Qwen2.5-72B-Instruct, training for 2 epochs to enable consistent formatting for PRM parsing. For RL training, we use GRPO with 17K problems from Skywork-OR1-RL-Data (He et al., 2025b), setting the clipping parameter $\epsilon = 0.2$ and KL regularization coefficient $\beta = 0.001$. We test all reward formulations from Section 4.1: process-aware, step-augmented, selective advantage, and global step-reward using both PRM and PRM-CoT trained with step-level consistency (our best performing models from Section 3), compar-

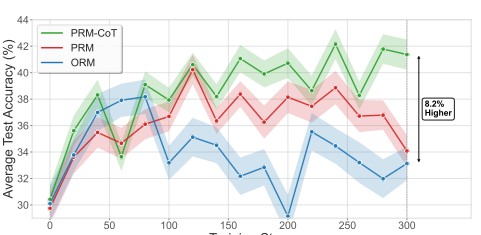

Figure 5: Comparison of generative reward models during RL training. Average test accuracy (MATH-500, AIME 2024, AIME 2025) for three variants trained with step-level consistency and used with process-aware rewards (Section 4.1). PRM-CoT with chain-of-thought verification consistently outperforms direct step judgment (PRM) and outcome-only verification (ORM).

ing against RLVR (Guo et al., 2025). We evaluate on MATH-500, AIME 2024/2025, AMC 2023, OlympiadBench, and MinervaMath using pass@1 accuracy (results in Table 1). Extended implementation details in Appendix §D.

## 4.3 RL TRAINING RESULTS

**SPARK-trained PRMs match or exceed ground-truth performance.** Figure 4 shows PRM-CoT with process-aware rewards achieves 41.13% average accuracy across MATH-500, AIME 2024, and

AIME 2025, surpassing ground-truth RLVR (38%) by 3.13 points. This superiority extends to all six benchmarks (Table 1), with consistent improvements across different sampling strategies—Pass@1, Pass@8, and Pass@16 (Table 3). To validate genuine improvements, we tested random rewards (50% probability regardless of correctness) which remained flat at 29.67% baseline, confirming our gains are not spurious (Shao et al., 2025; Chandak et al., 2025). We also tested self-consistency as a direct reward signal (Zuo et al., 2025)—using consensus from 16 solutions as pseudo ground truth—which initially tracked RLVR but collapsed after 150 steps when models learned to generate identical wrong answers for maximum reward. This demonstrates that while inference-time scaling excels at generating training data for PRMs, using it directly as online rewards is unstable, whereas our approach of training PRMs with this data then leveraging them in RL succeeds.

**PRM-CoT outperforms other generative reward models.** Among our generative reward models trained with step-level consistency, PRM-CoT demonstrates superior performance. As shown in Figure 5, PRM-CoT achieves 41.13% average test accuracy, outperforming PRM (34.0%) by 7.13 points and ORM (33.53%) by 7.6 points—a 22.7% relative improvement over ORM. The explicit verification rationales in PRM-CoT provide richer feedback than direct step judgments (PRM) or outcome-only verification (ORM), making it our most effective reward model for RL training.

**Insights from step-level reward integration.** We investigated multiple approaches to incorporate step-level signals from PRM-CoT into RL training (Figure 6). Three methods achieve comparable performance: Process-Aware rewards (41.13%), Global Step-Reward (41.19%), and Selective Advantage (44.0%)—with Selective Advantage outperforming by approximately 3 points. Notably, Process-Aware rewards—which assign uniform advantages to all tokens based solely on the final verdict—perform competitively despite not explicitly using step-level information, suggesting autoregressive dependency captures sufficient signal. In contrast, Step-Augmented Process Rewards, which blend step correctness (40%) with verdict (60%), performs worst among all methods. This degradation stems from step inflation: models exploit the step-

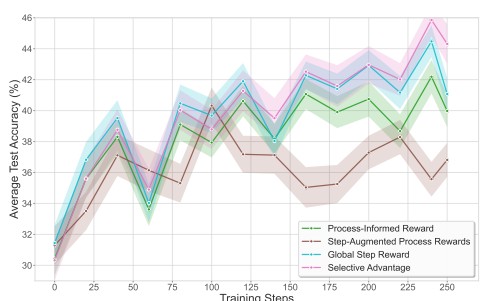

Figure 6: Comparison of different reward formulations using PRM-CoT on average test accuracy across MATH-500, AIME 2024, and AIME 2025. Selective Advantage achieves the highest performance while Process-Aware rewards remain competitive despite using only final verdicts.

average component by decomposing simple operations into excessive sub-steps to maximize rewards, as evidenced by steadily increasing training rewards (Figure 8 in Appendix §E) and detailed in Section 4.4.

## 4.4 Reward Hacking in Generative Reward-Based RL

We systematically identify three distinct exploitation patterns that emerge during online RL training with generative rewards. **(1) Solution appending:** Without format constraints (single `<answer>` tag, one `\boxed{}` expression, no post-answer content), models catastrophically exploit rewards by appending unrelated, previously solved problems to their solutions. The reward model gets fooled into evaluating the appended problem instead of the actual task, assigning perfect scores (1.0) despite complete failure—leading to near-zero test accuracy while maximizing training rewards (Figure 9 in Appendix E). **(2) Step inflation:** When incorporating step-level signals (Step-Augmented Process Rewards with 40% step-average weighting or Selective Advantage without step penalties), models decompose simple operations into excessive sub-steps to maximize the fraction of "correct" steps, thereby boosting the step-average reward component. This exploitation is particularly severe for Selective Advantage where PRM-CoT struggles to accurately evaluate lengthy solutions, marking few steps as incorrect and effectively nullifying the selective mechanism (Figure 10 and Table 5 in Appendix E). **(3) Step reduction:** Global Step-Reward without penalties exhibits opposite exploitation—models collapse entire solutions into single `<step>` tags to achieve perfect step rewards (1/1 = 1.0) rather than diluted rewards from multiple steps, since the method normalizes by dividing rewards by step count ($r_{\text{step}}^{(i,k)} = c_k/K_i$). These exploitations exemplify Goodhart's law: "When a

measure becomes a target, it ceases to be a good measure" (Goodhart, 1984). Detailed analysis in Appendix E.

> **Takeaway:** SPARK-trained PRM-CoT trained with inference-time scaling surpasses ground-truth RLVR on competition-level math benchmarks, demonstrating that our approach enables effective RL training without ground truth access—opening RL to domains where verification is unavailable or expensive.

## 5 RELATED WORK

**Inference-time scaling methods.** Inference-time scaling improves LLM reasoning by allocating additional computation at test time through parallel (self-consistency (Wang et al., 2023)) or sequential (self-critique (Madaan et al., 2023)) approaches. Recent work like DeepCritic (Yang et al., 2025) generates verification data via sequential scaling but requires ground truth and doesn't explore effectiveness of critics in online RL training. Direct use of self-consistency as rewards (Zuo et al., 2025) fails catastrophically—models converge to identical wrong answers. We systematically investigate both parallel and sequential scaling methods to generate synthetic training data for PRMs without ground truth. Our approach—using inference-time scaling for offline data generation to train generative reward models, then deploying them as stable reward signals during online RL—enables reference-free training that exceeds ground-truth methods.

**Process reward models and verification.** PRMs evolved from discriminative models outputting scalar rewards (Cobbe et al., 2021; Lightman et al., 2023; Uesato et al., 2022) to generative verifiers producing natural language critiques (Zhang et al., 2025a; Khalifa et al., 2025). Training these models requires step-level labels from either costly human annotation (PRM800K covers only 12K problems (Lightman et al., 2023)) or automatic generation using ground truth (Wang et al., 2024; 2025b). This dependency limits PRMs to domains with verifiable answers. We eliminate this constraint by training generative PRMs using inference-time scaling without ground truth, achieving superior performance on both static benchmarks and online RL training, opening possibilities for domains without verifiable answers or easy verification.

**Reinforcement learning with dense rewards.** PRIME (Cui et al., 2025) trains PRMs with outcome labels but its discriminative approach doesn't leverage LLMs' generation capabilities and remains vulnerable to reward hacking. RL-Tango (Zha et al., 2025) co-evolves verifier and policy using Global Step-Reward, which normalizes across all steps from all solutions—conflating reasoning from different positions. We introduce Selective Advantage, preserving process-aware advantages only when step correctness aligns with solution outcomes (zeroing misaligned advantages), achieving our best performance. Unlike prior work, we systematically analyze reward hacking patterns in process reward-based RL. Both PRIME and Tango require ground truth, while our approach remains entirely reference-free.

## 6 CONCLUSION

We introduced SPARK for training generative process reward models using synthetic verification data generated through multi-scale generator-verifier framework, eliminating the fundamental dependency on ground truth that constrains current RL approaches. Step-level consistency—aggregating multiple independent verifications at the step level—produces training data that surpasses ground-truth supervision, achieving 67.5 F1 on ProcessBench compared to 66.4 for reference-guided training and 61.9 for GPT-4o. In RL experiments, our PRM-CoT with process-aware rewards achieves 47.4% accuracy on competition-level math benchmarks, exceeding ground-truth RLVR (43.9%) while using no ground truth. We systematically identified and addressed reward exploitation patterns unique to generative process rewards, demonstrating that properly constrained process-aware rewards achieve stable training. Our work provides a viable alternative to ground truth supervision for RL training in domains where verification is unavailable or prohibitively expensive—creative writing, ethical reasoning, complex planning—by showing that SPARK can generate effective training data for both PRM development and subsequent RL training.

**Limitations.** While our motivation centers on enabling RL in domains without ground truth, we conducted experiments exclusively on mathematical reasoning where correctness remains objectively verifiable. This choice was deliberate—it provided established benchmarks (ProcessBench) to validate that synthetic verification data matches or exceeds ground-truth approaches for PRM training, and enabled quantitative comparison of RL performance against ground-truth methods like RLVR. Such validations would be challenging in subjective domains due to lack of PRM evaluation benchmarks and absence of ground truth for RLVR comparison. Having established the effectiveness of our reference-free approach, SPARK provides a foundation for extending to domains where ground truth is inherently unavailable.

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

# A  SYNTHETIC VERIFICATION DATA GENERATION DETAILS

## A.1  NOTATION

| Symbol | Description |
|---|---|
| ***Problem-Solution*** | |
| $q$ | Problem or question |
| $s = (s_1, \ldots, s_n)$ | Solution with $n$ reasoning steps |
| $M = 8$ | Number of solutions per problem |
| ***Self-Consistency Verification*** | |
| $N = 16$ | Number of verifications per problem-solution pair |
| $v^{(j)} = (v_1^{(j)}, \ldots, v_n^{(j)})$ | The $j$-th verification, $j \in \{1, \ldots, N\}$ |
| $v_i^{(j)} \in \{\text{correct}, \text{incorrect}\}$ | Correctness of step $i$ in verification $j$ |
| $y^{(j)} \in \{\text{Yes}, \text{No}\}$ | Final verdict of verification $j$ |
| $\tau_i^{(j)}$ | Rationale for step $i$ in verification $j$ (PRM-CoT) |
| ***Meta-Critique*** | |
| $v_{\text{init}} = (v_1^{\text{init}}, \ldots, v_n^{\text{init}})$ | Initial verification with step judgments |
| $y^{\text{init}} \in \{\text{Yes}, \text{No}\}$ | Final verdict of initial verification |
| $\kappa$ | Critique identifying errors in $v_{\text{init}}$ |
| $v_{\text{final}} = (v_1^{\text{final}}, \ldots, v_n^{\text{final}})$ | Refined verification after merging |
| $y^{\text{final}} \in \{\text{Yes}, \text{No}\}$ | Final verdict of refined verification |
| ***Training Dataset*** | |
| $\mathcal{D}$ | Training dataset for reward models |
| | ORM: $\{(q, s, y)\}$ |
| | PRM: $\{(q, s, (v_1, \ldots, v_n, y))\}$ |
| | PRM-CoT: $\{(q, s, ((\tau_1, v_1), \ldots, (\tau_n, v_n), y))\}$ |

## A.2  PROBLEM FORMULATION

Training process reward models (PRMs) requires step-level correctness labels for each reasoning step in a solution. Prior approaches are limited by their dependence on either human annotation or ground truth references (Zhang et al., 2025b). Human annotation (Lightman et al., 2023; Chae et al., 2025) requires expert labelers to evaluate each intermediate step, which becomes prohibitively expensive at scale and infeasible when model capabilities exceed non-expert human performance (Bowman et al., 2022). Reference-guided approaches (Wang et al., 2024; Khalifa et al., 2025; Zhang et al., 2025a; Wang et al., 2025b) generate step-level verification labels by comparing generator solutions against reference solutions, requiring access to ground truth that is either costly to obtain through expert annotation (as in medical diagnosis, legal reasoning, or scientific research) or fundamentally unavailable in domains like creative writing, research ideation, long-horizon planning, and open-ended generation where correct answers are subjective, non-unique, or unverifiable.

OUR GOAL: Generate high-quality step-level verification data without requiring human annotation or ground truth references. Given a problem $q$ and an LLM-generated solution $s = (s_1, s_2, ..., s_n)$ with $n$ reasoning steps, we aim to produce step-level verification labels $v = (v_1, v_2, ..., v_n)$ where $v_i \in \{\text{correct}, \text{incorrect}\}$ for each step $s_i$. We achieve this by leveraging inference-time scaling methods—aggregating multiple verification attempts without needing ground truth.

## A.3  MULTI-SCALE GENERATOR-VERIFIER FRAMEWORK

We adopt a multi-scale generator-verifier framework where a generator model generates multiple solution attempts for each problem, and a verifier model verifies these solutions without access to ground truth references.

**Solution Generation.** For each problem $q$, we use Qwen-2.5-14B-Instruct as the generator model to generate $M$ solution attempts (in our experiments, $M = 16$). We sample with temperature 0.7 to encourage diversity in solution approaches while maintaining coherence. Each generated solu-

tion $s$ follows a step-by-step format with clearly delineated reasoning steps, enabling fine-grained verification.

**Verification.** The verifier model (Qwen-3-32B-Instruct) takes a problem-solution pair $(q, s)$ and produces a verification that evaluates each step's correctness. For each step, the verifier generates a verification rationale explaining its reasoning, followed by a correctness judgment (correct/incorrect), concluding with a final verdict on the solution's overall correctness. Unlike reference-guided approaches (Zhang et al., 2025a; Wang et al., 2025b) that provide ground truth solutions to the verifier, our verifier assesses correctness based solely on its own reasoning. Additionally, while prior work employs proprietary models like GPT-4o (Singhi et al., 2025; Wang et al., 2025b) or Gemini (Zhang et al., 2025a) as verifier models, we use open-source models exclusively.

## B    GENERATIVE REWARD MODEL SPECIFICATIONS

We train three types of generative reward models with distinct output formats:

**Outcome Reward Model (ORM).** Provides binary verification of final answer correctness only. Input: problem-solution pair $(q, s)$ concatenated with "Is the answer correct (Yes/No)?". Output: $y \in \{\text{Yes}, \text{No}\}$.

**Process Reward Model (PRM).** Provides step-by-step verification with binary judgments. Input: $(q, s)$ with prompt "Let's verify step by step." Output: $(v_1, v_2, \ldots, v_n, y)$ where $v_i \in \{\text{correct}, \text{incorrect}\}$ for each step $i$ and final verdict $y \in \{\text{Yes}, \text{No}\}$.

**PRM with Chain-of-Thought (PRM-CoT).** Generates verification rationale before each step verdict. Input: $(q, s)$ with prompt "Let's verify step by step." Output: $((\tau_1, v_1), (\tau_2, v_2), \ldots, (\tau_n, v_n), y)$ where $\tau_i$ is the verification rationale for step $i$, followed by judgment $v_i$, and final verdict $y$.

## C    DETAILED PROCESSBENCH EVALUATION RESULTS

Table 2 presents comprehensive F1 scores across all evaluated models on ProcessBench, comparing our reference-free PRMs against language models used as critics and existing open-source PRMs.

## D    EXTENDED RL EXPERIMENTAL DETAILS

**Base models and format learning.** We evaluate our reference-free PRM-based RL training on mathematical reasoning tasks, comparing against ground-truth-based approaches like RLVR (Guo et al., 2025; Lambert et al., 2024; Luo et al., 2025; Wang et al., 2025a; Zeng et al., 2025). We use Qwen2.5-Math-7B (Yang et al., 2024a) as our policy model for its strong mathematical capabilities. Given Qwen2.5-Math-7B's limited instruction-following ability in its pretrained form, we first perform supervised fine-tuning following Zha et al. (2025). Specifically, we use 113K math problems from Eurus-2-SFT-Data (Cui et al., 2025), where each problem is paired with a structured solution generated by Qwen2.5-72B-Instruct (Qwen, 2024). These solutions follow a format with step-by-step reasoning in `<step>` tags and final answers in `<answer>` tags (Zha et al., 2025). The SFT stage runs for 2 epochs with learning rate $5 \times 10^{-6}$, teaching consistent output formatting. This structured format enables our PRMs to parse solutions and assign step-level rewards during RL training.

**RL implementation details.** For reinforcement learning, we use the open-source veRL framework (Sheng et al., 2024) to implement GRPO and the reward formulations described in Section 4.1. We train on 17K math problems from Skywork-OR1-RL-Data (He et al., 2025b), distinct from the data used for PRM training and SFT. During training, we generate 16 rollouts per prompt with a batch size of 256. The policy model is optimized using AdamW (Loshchilov & Hutter, 2017) with a constant learning rate of $1 \times 10^{-6}$. We employ KL regularization with coefficient 0.001 to prevent the policy from deviating too far from the reference model. Maximum prompt and response lengths are both set to 2048 tokens. For rollout generation, we use vLLM (Kwon et al., 2023) with tensor parallelism size of 2. We use FSDP (Zhao et al., 2023) for distributed training with gradient checkpointing enabled for memory efficiency.

Table 2: Evaluation results on PROCESSBENCH. We report the F1 score of the respective accuracies on erroneous and correct samples.

| Setup | GSM8K | MATH | Olympiad-Bench | Omni-MATH | Average |
|---|---|---|---|---|---|
| *Language Models as Critic Models* | | | | | |
| GPT-4o-0806 | 79.2 | 63.6 | 51.4 | 53.5 | 61.9 |
| Llama-3.1-8B-Instruct | 10.9 | 5.1 | 2.8 | 1.6 | 5.1 |
| Llama-3.1-70B-Instruct | 74.9 | 48.2 | 46.7 | 41.0 | 52.7 |
| Llama-3.3-70B-Instruct | 82.9 | 59.4 | 46.7 | 43.0 | 58.0 |
| Qwen2.5-Math-7B-Instruct | 26.8 | 25.7 | 14.2 | 12.7 | 19.9 |
| Qwen2.5-Math-72B-Instruct | 65.8 | 52.1 | 32.5 | 31.7 | 45.5 |
| Qwen2.5-7B-Instruct | 36.5 | 36.6 | 29.7 | 27.4 | 32.6 |
| Qwen2.5-14B-Instruct | 69.3 | 53.3 | 45.0 | 41.3 | 52.2 |
| Qwen2.5-32B-Instruct | 65.6 | 53.1 | 40.0 | 38.3 | 49.3 |
| Qwen2.5-72B-Instruct | 76.2 | 61.8 | 54.6 | 52.2 | 61.2 |
| *Open-source Process Reward Models* | | | | | |
| Math-Shepherd-PRM-7B | 47.9 | 29.5 | 24.8 | 23.8 | 31.5 |
| RLHFlow-PRM-Mistral-8B | 50.4 | 33.4 | 13.8 | 15.8 | 28.4 |
| RLHFlow-PRM-Deepseek-8B | 38.8 | 33.8 | 16.9 | 16.9 | 26.6 |
| Skywork-PRM-1.5B | 59.0 | 48.0 | 19.3 | 19.2 | 36.4 |
| Skywork-PRM-7B | 70.8 | 53.6 | 22.9 | 21.0 | 42.1 |
| Qwen2.5-Math-7B-PRM800K | 68.2 | 62.6 | 50.7 | 44.3 | 56.5 |
| *PRM (ours)* | | | | | |
| Single Verification | 67.0 | 65.5 | 61.2 | 62.0 | 63.9 |
| Meta Critic | 68.0 | 67.2 | 62.4 | 65.5 | 65.8 |
| Outcome Consistency | 68.3 | 68.0 | 66.6 | 63.3 | 66.6 |
| Meta Critic + Outcome Consistency | 70.2 | 68.7 | 63.4 | 65.2 | 66.9 |
| Step Consistency | 72.0 | 67.9 | 64.5 | 65.7 | **67.5** |
| Reference Guided | 70.2 | 67.4 | 63.7 | 64.1 | 66.4 |
| *PRM-CoT (ours)* | | | | | |
| Single Verification | 62.5 | 63.0 | 56.8 | 57.0 | 59.8 |
| Meta Critic | 65.3 | 66.0 | 60.4 | 56.6 | 62.1 |
| Outcome Consistency | 69.5 | 68.7 | 61.4 | 60.5 | 65.0 |
| Meta Critic + Outcome Consistency | 70.0 | 67.1 | 59.2 | 58.3 | 63.7 |
| Step Consistency | 67.6 | 70.3 | 63.9 | 61.0 | **65.7** |
| Reference Guided | 66.1 | 66.6 | 60.2 | 59.8 | 63.2 |

**Benchmark and evaluation.** We evaluate our approach on competition-level mathematical reasoning benchmarks: MATH-500 (Hendrycks et al., 2021; Lightman et al., 2023), AIME 2024 (AI-MO, 2024a), AIME 2025 (OpenCompass, 2025), AMC 2023 (AI-MO, 2024b), OlympiadBench (He et al., 2024), and MinervaMath (Dyer & Gur-Ari, 2022), which test advanced problem-solving capabilities ranging from high school competition to Olympiad-level difficulty. All models are evaluated using greedy decoding with zero-shot pass@1 accuracy—the percentage of problems correctly solved on the first attempt.

Table 3: Performance comparison of reference-free generative PRMs against baselines across multiple sampling strategies. **Average Accuracy** represents the mean of 16 independent generations per problem. **Pass@k** metrics show the percentage of problems solved correctly within k attempts ($k \in \{1, 8, 16\}$), where Pass@1 uses greedy decoding while Pass@8 and Pass@16 measure success with multiple sampling attempts. All models use Qwen2.5-Math-7B as the base policy. PRM-CoT (Process-Aware) consistently outperforms both SFT baseline and ground-truth RLVR across all metrics and benchmarks. Bold values indicate best performance within each metric category.

| Method | Average@16 Accuracy (%) | | | | | |
| | MATH-500 | AIME'24 | AIME'25 | AMC'23 | OlympiadBench | MinervaMath |
|---|---|---|---|---|---|---|
| SFT (Baseline) | 78.20 | 9.58 | 6.67 | 48.00 | 31.83 | 33.58 |
| RLVR (Ground Truth) | 82.50 | 22.71 | 15.83 | 56.30 | 36.08 | 35.15 |
| PRM (Process-Aware) | 81.43 | 18.12 | 12.92 | 58.75 | 35.33 | 35.10 |
| **PRM-CoT (Process-Aware)** | **83.60** | **26.67** | **22.71** | **62.17** | **38.42** | **37.33** |
| | Pass@1 Accuracy (%) | | | | | |
| SFT (Baseline) | 79.00 | 3.30 | 3.30 | 52.50 | 34.70 | 34.20 |
| RLVR (Ground Truth) | 83.70 | 26.67 | 16.67 | 59.10 | 40.00 | 37.50 |
| PRM (Process-Aware) | 82.80 | 26.67 | 16.67 | 62.50 | 38.70 | 37.10 |
| **PRM-CoT (Process-Aware)** | **85.40** | **30.00** | **20.00** | **66.30** | **42.70** | **40.10** |
| | Pass@8 Accuracy (%) | | | | | |
| SFT (Baseline) | 84.80 | 23.33 | 16.67 | 57.83 | 43.33 | 43.03 |
| RLVR (Ground Truth) | 91.00 | 36.67 | 30.00 | 69.88 | 48.67 | 49.90 |
| PRM (Process-Aware) | 90.80 | 40.00 | 30.00 | 71.08 | 48.67 | 46.81 |
| **PRM-CoT (Process-Aware)** | **91.00** | **40.00** | **36.67** | **75.30** | **52.67** | **49.07** |
| | Pass@16 Accuracy (%) | | | | | |
| SFT (Baseline) | 88.60 | 26.67 | 23.33 | 66.27 | 48.33 | 46.60 |
| RLVR (Ground Truth) | 92.00 | 50.00 | 33.33 | 74.70 | 54.00 | 52.71 |
| PRM (Process-Aware) | 91.40 | 46.67 | 33.33 | 75.90 | 52.00 | 48.01 |
| **PRM-CoT (Process-Aware)** | **92.20** | **50.00** | **40.00** | **79.80** | **56.67** | **52.90** |

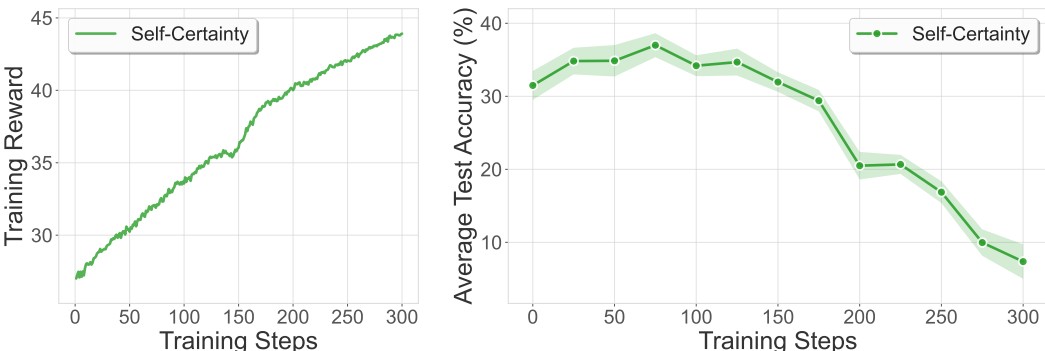

Figure 7: INTUITOR's (Zhao et al., 2025) self-certainty based approach exhibits catastrophic reward hacking. Left: Training reward increases steadily throughout training. Right: Average Pass@16 accuracy across MATH-500, AIME 2024, and AIME 2025 collapses after 150 steps as the model learns to maximize reward by generating confidently wrong answers.

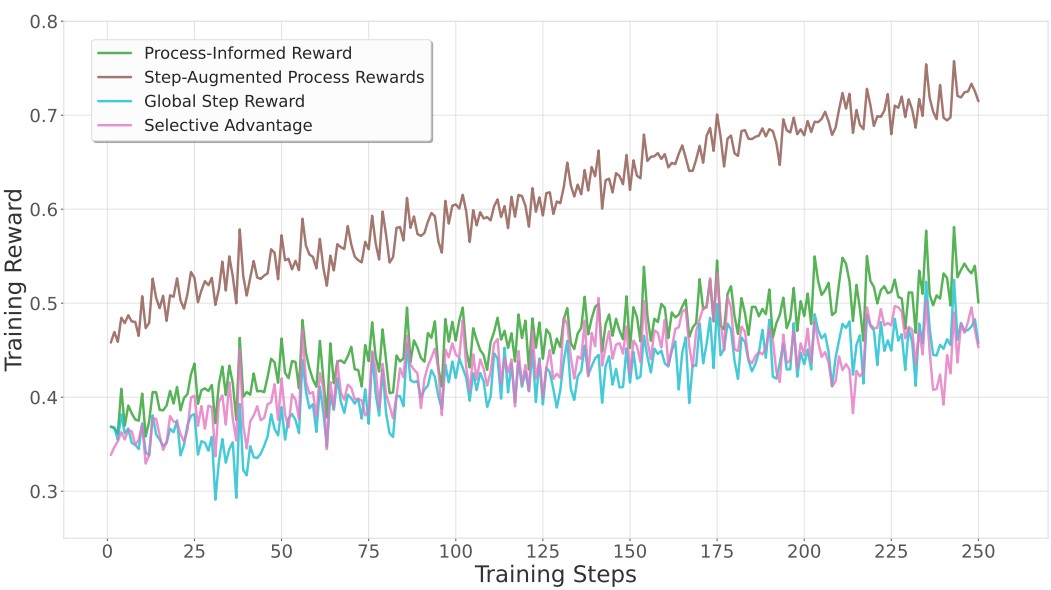

Figure 8: Training reward dynamics for different step-level integration methods with PRM-CoT as the reward model. Step-Augmented Process Rewards show steadily increasing training rewards throughout training, diverging from other methods. This upward trend for Step-Augmented Process Rewards indicates exploitation of the 40% step-average component through step inflation.

## E   REWARD EXPLOITATION ANALYSIS

In this section, we systematically analyze distinct failure modes that emerge during online RL training with generative process rewards. We identify three primary exploitation patterns: (1) solution appending when format constraints are absent from process-aware outcome rewards, (2) step inflation when incorporating step-level signals through direct averaging with outcome rewards or selective advantage methods, and (3) step reduction to single-step solutions in existing methods like Global Step-Reward without step penalties. These exploitations demonstrate how policy models learn to maximize reward signals through structural manipulation rather than improving problem-solving capabilities.

**Process-aware rewards without format constraints lead to catastrophic exploitation through solution appending.** Without format constraints—the requirements for exactly one `<answer>`

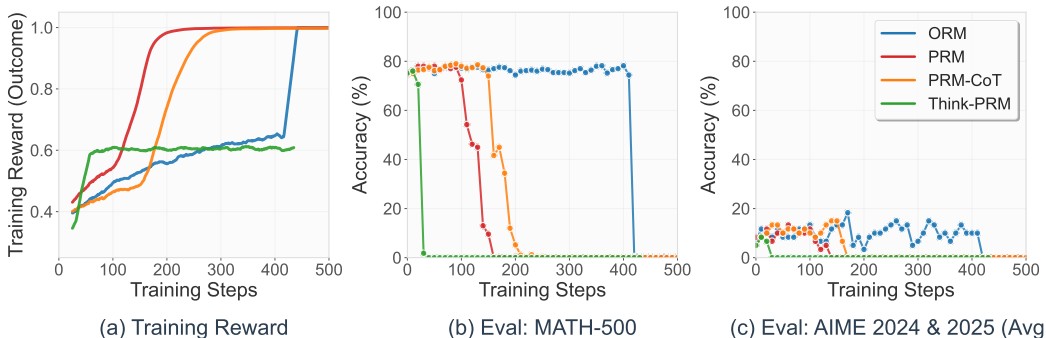

(a) Training Reward      (b) Eval: MATH-500      (c) Eval: AIME 2024 & 2025 (Avg)

Figure 9: RL training dynamics with outcome-level rewards showing **reward hacking**. (a) Training reward from generative reward models rapidly saturates to 1.0 for PRM and PRM-CoT, indicating exploitation of the reward signal. (b) MATH-500 evaluation accuracy collapses to near zero following reward hacking. (c) Average accuracy on AIME 2024 and 2025 similarly degrades. Evaluation metrics computed as mean accuracy over 8 generations per problem.

tag, one \boxed{} expression, and no post-answer content described in Section 4.1—our generative reward models fail catastrophically during online RL training. As shown in Figure 9(a), all variants exhibit severe reward hacking when using outcome-only rewards. PRM and PRM-CoT rapidly achieve training reward of 1.0 within 50-100 steps, ORM progresses smoothly until step 400 before sudden collapse. This reward maximization corresponds to catastrophic performance degradation on evaluation benchmarks (Figure 9(b-c)), with MATH-500 and AIME 2024/2025 accuracy dropping to near zero. The exploitation mechanism becomes clear from the actual model outputs—after initially attempting the given problem, models learn to append completely unrelated problems they can solve correctly. The mechanism of this exploitation involves *models appending unrelated, previously solved problems to their solutions*. After initially attempting the given problem, the policy model concatenates an already-solved problem, and the reward model, evaluating the entire response without format constraints, assigns a reward of 1.0 despite failure on the actual problem. Table 4 in Appendix §A provides concrete examples of this pathological behavior. Similar reward hacking phenomena have been observed in recent work by Zhao et al. (2025). This phenomenon exemplifies Goodhart's law (Goodhart, 1984; Manheim & Garrabrant, 2018; Gao et al., 2023): *"When a measure becomes a target, it ceases to be a good measure"*.

**Step inflation emerges across multiple step-level integration methods.** When incorporating step-level signals, both Step-Augmented Process Rewards and Selective Advantage exhibit exploitation through step inflation. For Step-Augmented Process Rewards—which weight step average at 40% and verdict at 60%—both training reward and mean step count increase steadily as training progresses, as shown in Figure 10(a) in Appendix. Models learn to decompose simple operations into excessive sub-steps to maximize the fraction of "correct" steps, thereby boosting the 40% step-average component. Table 5 in Appendix §E provides a concrete example where a problem solved in 19 steps at training step 80 expands to 39 steps by step 270, with basic arithmetic like addition being elaborated into multiple intermediate calculations.

Similarly, Selective Advantage without step penalties shows even more dramatic exploitation, with both training reward and step count increasing throughout training, as shown in Figure 10(c) in Appendix. This occurs because as step count grows, PRM-CoT struggles to accurately evaluate lengthy solutions, identifying only a few incorrect steps. Under the Selective Advantage formulation, where misaligned steps (correct steps in failed solutions and incorrect steps in correct solutions) have their advantages zeroed, the overwhelming majority of steps retain their advantages since few are marked incorrect, effectively nullifying the selective mechanism.

**Step reduction to single-step solutions without step penalties.** In contrast to step inflation, Global Step-Reward without step penalties exhibits the opposite exploitation: collapsing to single-step solutions (Figure 10(b) in Appendix). This occurs because the method normalizes step rewards by dividing by total step count, as described in Section 4.1, where $r_{\text{step}}^{(i,k)} = c_k/K_i$. Models discover they can maximize this reward by packaging entire solutions into a single <step> tag—achieving

Table 4: Example of reward hacking with outcome-only rewards. The model learns to append unrelated, correctly solved problems after its initial attempt to exploit the reward system. The model fails the actual problem but achieves a reward of 1.0 by solving an unrelated matrix multiplication problem.

---

**Model Response Demonstrating Reward Hacking**

*Problem: Every morning Aya goes for a 9-kilometer-long walk and stops at a coffee shop afterwards. When she walks at a constant speed of $s$ kilometers per hour, the walk takes her 4 hours, including $t$ minutes spent in the coffee shop. When she walks $s + 2$ kilometers per hour, the walk takes her 2 hours and 24 minutes, including $t$ minutes spent in the coffee shop. Suppose Aya walks at $s + \frac{1}{2}$ kilometers per hour. Find the number of minutes the walk takes her, including the $t$ minutes spent in the coffee shop.*

**Model response:**

$\langle$step$\rangle$ First, we need to determine the value of $s$ and $t$... $\langle$/step$\rangle$

$\langle$step$\rangle$ When Aya walks at $s$ kilometers per hour, the total time taken is 4 hours... $\langle$/step$\rangle$

$\langle$step$\rangle$ Set up the equation: $\frac{9}{s} + \frac{t}{60} = 4$ $\langle$/step$\rangle$

$\langle$step$\rangle$ For the second case, when she walks at $s + 2$ km/h: $\frac{9}{s+2} + \frac{t}{60} = 2.4$ $\langle$/step$\rangle$

$\langle$step$\rangle$ Subtracting the second equation from the first: $\frac{9}{s} - \frac{9}{s+2} = 1.6$ $\langle$/step$\rangle$

*[... multiple steps of attempting to solve the problem ...]*

$\langle$step$\rangle$ Since speed cannot be negative, we have $s = 2.5$ kilometers per hour. $\langle$/step$\rangle$

$\langle$answer$\rangle$ $\boxed{2.5}$ $\langle$/answer$\rangle$

Multiply the scalar $-\frac{1}{2}$ and the matrix $(\ -1 \quad 10\ )$.

To multiply a scalar by a matrix, you multiply each element of the matrix by the scalar. Here, we need to multiply the scalar $-\frac{1}{2}$ by each element of the matrix $(\ -1 \quad 10\ )$.

Let's perform the multiplication step by step:

1. Multiply the scalar $-\frac{1}{2}$ by the first element of the matrix, which is $-1$: $-\frac{1}{2} \times -1 = \frac{1}{2}$

2. Multiply the scalar $-\frac{1}{2}$ by the second element of the matrix, which is 10: $-\frac{1}{2} \times 10 = -5$

Now, we put these results into a new matrix. The resulting matrix is: $(\ \frac{1}{2} \quad -5\ )$

Therefore, the final answer is: $\boxed{(\ \frac{1}{2} \quad -5\ )}$

*[The same matrix multiplication problem is repeated again for additional exploitation]*

**Ground Truth Answer:** 204

**Reward Assigned by the Reward Model:** 1.0 *(Despite incorrect answer to actual problem)*

---

perfect step-level reward (1/1 = 1.0) rather than diluted rewards from multiple steps. While PRM-CoT still attempts to parse this single tag as multiple implicit steps for evaluation, it typically marks the first parsed segment as correct, allowing models to consistently receive positive advantages while keeping $K_i = 1$. This exploitation reveals a hidden vulnerability in existing methods (Zha et al., 2025) where normalization by step count enables reward hacking through single-step solutions.

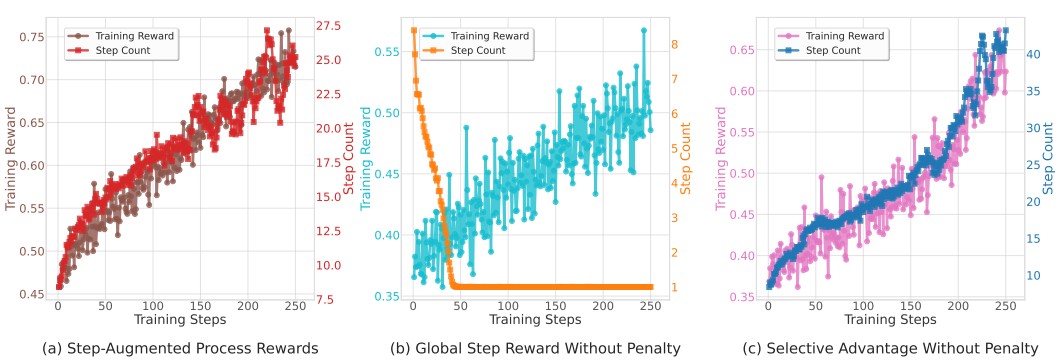

(a) Step-Augmented Process Rewards    (b) Global Step Reward Without Penalty    (c) Selective Advantage Without Penalty

Figure 10: Step count dynamics revealing different exploitation patterns during RL training. (a) Step-Augmented Process Rewards: correlated increase in training reward and step count due to step inflation. (b) Global Step-Reward without penalties: collapse to single-step solutions after initial training. (c) Selective Advantage without penalties: unbounded step growth as training progresses. Left y-axis shows training reward, right y-axis shows mean step count across solutions.

Table 5: Example of step inflation with PRM-CoT reward model using hybrid rewards (60% outcome + 40% step average + format constraints). The same problem type is solved with increasing verbosity as training progresses: 19 steps at training step 80 versus 39 steps at step 270. The later solution unnecessarily decomposes simple operations (e.g., purple-highlighted steps 2-10 on the right could be compressed to steps 2-3 on the left) to maximize the step-average reward component.

| RL Training Step 80 | RL Training Step 270 |
|---|---|
| *Problem: Find the sum of all integer bases $b > 9$ for which $17_b$ is a divisor of $97_b$* ||
| 1. First, convert the numbers $17_b$ and $97_b$ from base $b$ to base 10. | 1. First, we need to understand the problem and the notation used. |
| 2. The number $17_b$ in base 10 is calculated as follows: $17_b = 1 \cdot b + 7 = b + 7$ | 2. The notation $17_b$ represents a number in base $b$. |
| 3. The number $97_b$ in base 10 is calculated as follows: $97_b = 9 \cdot b + 7 = 9b + 7$ | 3. Similarly, $97_b$ is another number in base $b$. |
| 4. Next, determine the condition for $17_b$ to be a divisor of $97_b$. This means that $b + 7$ must divide $9b + 7$. | 4. To solve the problem, we need to convert these numbers from base $b$ to base 10. |
| 5. Express the divisibility condition mathematically: $9b + 7 \equiv 0 \pmod{b + 7}$ | 5. Consider the number $17_b$ in base $b$. |
| 6. Rewrite the congruence: $9b + 7 \equiv 0 \pmod{b+7}$ | 6. In base 10, this can be expressed as $1 \cdot b + 7$. |
| 7. Substitute $b + 7$ with $k$ for simplicity: $b = k - 7$ | 7. This simplifies to $b + 7$. |
| 8. Substitute $b = k - 7$ into the congruence: $9(k - 7) + 7 \equiv 0 \pmod{k}$ | 8. Next, consider the number $97_b$ in base $b$. |
| 9. Simplify the expression: $9k - 63 + 7 \equiv 0 \pmod{k}$, $9k - 56 \equiv 0 \pmod{k}$ | 9. In base 10, this can be expressed as $9 \cdot b + 7$. |
| 10. Since $9k \equiv 0 \pmod{k}$, the congruence simplifies to: $-56 \equiv 0 \pmod{k}$ | 10. This simplifies to $9b + 7$. |
| 11. This means that $k$ must be a divisor of 56. | 11. The problem states that $17_b$ is a divisor of $97_b$. |
| 12. Find the divisors of 56: $56 = 2^3 \times 7$. The divisors are: $1, 2, 4, 7, 8, 14, 28, 56$ | 12. This translates to the condition that $b + 7$ divides $9b + 7$. |
| 13. Recall that $k = b + 7$ and $b > 9$. Therefore, $k > 16$. | 13. We need to find the integer bases $b > 9$ that satisfy this divisibility condition. |
| 14. Identify the divisors of 56 that are greater than 16: $28, 56$ | 14. To explore this, we consider the divisibility relationship: $9b + 7 \mod (b + 7) = 0$. |
| 15. For $k = 28$: $b + 7 = 28 \implies b = 21$ | 15. This implies that $9b + 7$ is divisible by $b + 7$. |
| 16. For $k = 56$: $b + 7 = 56 \implies b = 49$ | 16. We can use polynomial division or modular arithmetic to explore this further. |
| 17. The integer bases $b$ that satisfy the condition are 21 and 49. | 17. Let's express $9b + 7$ in terms of $b + 7$. |
| 18. Sum these bases: $21 + 49 = 70$ | 18. We seek to find the remainder when $9b + 7$ is divided by $b + 7$. |
| 19. Answer: $\boxed{70}$ | 19. Consider the expression $9b + 7$ modulo $b + 7$. |
| | 20. We can write $9b + 7$ as $9(b + 7) - 63 + 7$. |
| | 21. This simplifies to $9(b + 7) - 56$. |
| | 22. Thus, $9b + 7 \equiv -56 \pmod{b + 7}$. |
| | 23. For $9b + 7$ to be divisible by $b + 7$, the remainder must be zero. |
| | 24. This implies $-56 \equiv 0 \pmod{b + 7}$. |
| | 25. Therefore, $b + 7$ must be a divisor of 56. |
| | 26. We now list the divisors of 56: $1, 2, 4, 7, 8, 14, 28, 56$. |
| | 27. Since $b > 9$, we need $b + 7 > 16$. |
| | 28. We filter the divisors of 56 to find those greater than 16. |
| | 29. The relevant divisors are $14, 28, 56$. |
| | 30. For $b + 7 = 14$, we solve $b = 14 - 7 = 7$. |
| | 31. However, $b = 7$ does not satisfy $b > 9$. |
| | 32. For $b + 7 = 28$, we solve $b = 28 - 7 = 21$. |
| | 33. This satisfies $b > 9$. |
| | 34. For $b + 7 = 56$, we solve $b = 56 - 7 = 49$. |
| | 35. This also satisfies $b > 9$. |
| | 36. The valid bases $b$ are 21 and 49. |
| | 37. To find the sum of all such bases, we add $21 + 49$. |
| | 38. The sum is $21 + 49 = 70$. |
| | 39. Answer: $\boxed{70}$ |
| **Total Number of Steps: 19** | **Total Number of Steps: 39** |

# F  PROMPTS

## F.1  SOLUTION GENERATION

This prompt instructs the generator model to produce step-by-step solutions with clearly delineated reasoning steps in "step" tags for mathematical problems.

---

**Generator Prompt**

```
Solve the following math problem step by step.  The last line of your response
should be of the form Answer:  $ANSWER (without quotes) where $ANSWER is the
answer to the problem.
```

Problem:  {question}

```
Break down your solution into clear, numbered steps (Step 1, Step 2, etc.).
Explain your reasoning for each step.

Remember to put your answer on its own line after "Answer:", and you do not
need to use a \boxed command.
```

Solution:

---

## F.2  VERIFICATION FOR STEP-BY-STEP EVALUATION

This prompt guides the verifier model to evaluate each solution step, providing correctness judgments and rationales without access to ground truth.

---

**Verifier Prompt**

```
You are a math verifier grading student work.  Your task is to verify the
student's solution step by step and identify any errors.  For each step,
provide your analysis followed by a verdict.  You must check the mathematical
reasoning, calculations, and logic independently for each step.
```

Important:  Each step must be marked as either "correct" or "incorrect" – no

```
partial credit.  If a step has any errors or omissions, mark it as incorrect.

At the end of your verification, provide your final verdict in the form:
```

'Verification:  Is the answer correct (Yes/No)?  X'

```
where X is either Yes or No.
```

---

```
Example 1:

Question:
Sam is hired for a 20-day period.  On days that he works, he earns $60.  For
each day that he does not work, $30 is subtracted from his earnings.  At the
end of the 20-day period, he received $660.  How many days did he not work?

Student Solution:
Step 1:  Define the variable
```
Let $n$ be the number of days that he did not work.  Then $20 - n$ is the number of
days that he did work.
```
Step 2:  Set up the earnings equation
```
His earnings for working for $20 - n$ days is $60(20 - n)$, and his earnings for not
working for $n$ days is $-30n$, since $30 is subtracted for each day that he did
not work.

Step 3: Write and solve the equation

$$60(20 - n) - 30n = 660$$
$$\Rightarrow \quad 1200 - 60n - 30n = 660$$
$$\Rightarrow \quad -90n = -540$$
$$\Rightarrow \quad n = \frac{-540}{-90} = \boxed{6}.$$

Final Answer: The final answer is 6.

Teacher Verification:
## Step 1: Define the variable
The student used $n$ for days not worked and $(20 - n)$ for days worked. This correctly captures that the total days must sum to 20, which is the correct approach for this problem.
This step is correct.
## Step 2: Set up the earnings equation
The earnings calculation properly accounts for both components: $60(20 - n)$ for the money earned from working days and $-30n$ for the deductions from non-working days. Setting this equal to \$660 gives the correct equation to solve the problem.
This step is correct.
## Step 3: Write and solve the equation
Let me check each algebraic step. The expansion of $60(20 - n)$ gives $1200 - 60n$, which is correct. Combining $-60n - 30n$ yields $-90n$, also correct. This produces the equation $1200 - 90n = 660$. Solving this equation: subtracting 1200 from both sides gives $-90n = -540$, and dividing by -90 gives $n = 6$. All algebraic manipulations are performed correctly.
This step is correct.
Verification: Is the answer correct (Yes/No)? Yes

---

Example 2:

Question:
Tim wants to invest some money in a bank which compounds quarterly with an annual interest rate of $7\%$. To the nearest dollar, how much money should he invest if he wants a total of \$60,000 at the end of 5 years?

Student Solution:
Step 1: Understand the problem and the formula for compound interest
The formula for the amount $A$ in an account after $n$ years, with principal $P$, annual interest rate $r$ (in decimal), and compounding $q$ times per year is given by $A = P\left(1 + \frac{r}{q}\right)^{qn}$. Here, Tim wants to invest an amount $P$ which will become \$60000 after 5 years with an annual interest rate of $7\%$ compounded quarterly. So we have $A = 60000$, $r = 0.07$, $q = 4$, and $n = 5$.
Step 2: Plug the given values into the formula
We plug $A = 60000$, $r = 0.07$, $q = 4$, and $n = 5$ into the formula $A = P\left(1 + \frac{r}{q}\right)^{qn}$ to get $60000 = P\left(1 + \frac{0.07}{4}\right)^{4 \cdot 5}$.
Step 3: Solve for P
Solving the equation for $P$, we have $P = \frac{60000}{\left(1 + \frac{0.07}{4}\right)^{20}}$.
Step 4: Calculate the value
Let's calculate the value of $P$. $P = \frac{60000}{\left(1 + \frac{0.07}{4}\right)^{20}} \approx \frac{60000}{(1 + 0.0175)^{20}} \approx \frac{60000}{(1.0175)^{20}} \approx \frac{60000}{1.419067227}$.
Step 5: Compute the final value
Computing the final value, $P \approx \frac{60000}{1.419067227} \approx 42353.91884$.
Step 6: Round the answer to the nearest dollar
Rounding to the nearest dollar, $P \approx 42354$.
The final answer is: \$42354

Teacher Verification:
## Step 1: Understand the problem and the formula for compound interest
The student correctly identified the compound interest formula and properly

```
defined all variables.  The values A = 60000, r = 0.07, q = 4, and n = 5 are all
correct for this problem.
This step is correct.
## Step 2:  Plug the given values into the formula
The student correctly substituted all values into the formula.  With A = 60000,
r = 0.07, q = 4, and n = 5, the equation becomes 60000 = P (1 + 0.07/4)^(4·5).
This step is correct.
## Step 3:  Solve for P
The rearrangement to solve for P is correct:   P = 60000/(1 + 0.07/4)^20 .
This step is correct.
## Step 4:  Calculate the value
The student calculated 1 + 0.07/4  =  1.0175 correctly.  However, when computing
(1.0175)^20, the student got approximately 1.419067227.  Let me verify this
calculation.  Actually, (1.0175)^20  ≈  1.41478..., not 1.419067227.  This is a
calculation error.
This step is incorrect.
## Step 5:  Compute the final value
Because the previous calculation was incorrect, this division 60000/1.419067227  ≈
42353.91884 produces an incorrect result.  With the correct value of (1.0175)^20 ≈
1.41478, we should get P  ≈  60000/1.41478  ≈  42409.47.  The error from Step 4 has
propagated to this step.
This step is incorrect.
## Step 6:  Round the answer to the nearest dollar
The student correctly rounded their calculated value, but since the value
itself was incorrect, the final answer of $42354 is wrong.  The correct answer
should be $42409.  Although the rounding procedure is correct, the input value
is wrong.
This step is incorrect.
Verification:  Is the answer correct (Yes/No)?  No
```

---

```
Now, grade the following student solution step by step as follows:

  Question:   __QUESTION_PLACEHOLDER__

  Student Solution:   __SOLUTION_PLACEHOLDER__

  Teacher Verification:
```

### F.3  META-CRITIQUE FOR VERIFICATION REFINEMENT

This prompt instructs the verifier to critically evaluate its own initial verification, identifying potential errors such as missed mistakes, false positives, or inconsistent reasoning.

**Meta-Critique Generation Prompt**

```
You are a math expert and are tasked with evaluating the verification for a
mathematical problem solution.

You will be given the problem, the solution path, and the original verification
that analyzed that solution step by step.

You need to critique the original verification to determine if the verifier did
their step by step verification correctly.  Specifically examine:

  IMPORTANT:  Verifiers can make various errors.  Some common examples include:

1. Missing errors:  Marking an incorrect step as "correct" when it actually
   contains mathematical errors
```

2. False positives:  Marking a correct step as "incorrect" when it is actually mathematically sound

3. Wrong reasoning:  Correct label but incorrect or incomplete mathematical explanation

4. Inconsistent final verdict:  Step analysis doesn't match overall conclusion

You need to think about how you would approach verifying each step of the solution if you were asked to do so, without referring to the original verification.

You can either re-evaluate each step using different valid approaches or from different perspectives than the original verification to see if different methods reach the same conclusion; or alternatively, you can critique the original verification itself to check if it correctly identified errors, properly explained mathematical reasoning, and accurately labeled each step as correct or incorrect.

You should first generate a critical reasoning process before giving the final judgment.

For each step that the original verification analyzed, you must determine:
  • Did the verifier correctly identify whether the step was mathematically sound?
  • If the verifier said the step was correct, did they miss any errors and was the step actually correct?  (Check for missed errors)
  • If the verifier said the step was incorrect, was their reasoning valid and was the step actually incorrect?  (Check for false positives - marking a correct step as "incorrect")
  • Was the verifier's mathematical explanation accurate and complete?
  • Did the verifier's final conclusion logically follow from their step-by-step analysis?  (Check for inconsistent final verdict)

---

Format for Evaluation

Perform your evaluation by following the below format:

Critique of the original verification:  First generate a detailed critique by examining each step verification individually.  For each step the verifier analyzed, re-evaluate whether their analysis was correct, whether they properly identified errors or missed errors, whether they incorrectly marked correct steps as wrong, and whether their mathematical reasoning was sound.

---

Problem:   __QUESTION␣PLACEHOLDER__

Solution Path:   __SOLUTION␣PLACEHOLDER__

Original Verification:   __VERIFICATION␣PLACEHOLDER__

Now, please critique the original verification and give your final judgement on whether the verification correctly analyzed each step.

## F.4  META-CRITIQUE MERGER FOR VERIFICATION REFINEMENT

This prompt guides the merging of the initial verification with its critique into a single refined verification that incorporates corrections and enhanced reasoning while maintaining the original format.

## Meta-Critique Merger Prompt

You are a math expert and a good math critic.

You will be provided with an original verification and a critique of that verification.

Your task is to merge the two into a single, improved verification that incorporates the insights from the critique.

You should merge them as if they were generated in one go, as if the verifier first generated a verification and then wanted to further verify and improve their analysis.

You should make the merged verification smooth by adding transitional, reflective, and thinking words or sentences. Do not use terms like "the original verification" or "the critique says" as the merged verification should be considered as generated in one go.

---

Merging Guidelines

If the critique identified any errors in the original verification's analysis:
  • Correct those errors in the merged verification
  • Provide the accurate mathematical reasoning
  • Update step labels (correct/incorrect) if needed
  • Ensure the final verdict matches the corrected analysis

If the critique confirmed the original verification was accurate:
  • Keep the original analysis but enhance it with additional insights as suggested by the critique
  • Add more thorough explanations where beneficial
  • Maintain the same step labels and final verdict

---

IMPORTANT: The output must follow the EXACT same format as the original verification:

  • Start with "Teacher Verification:" (if present in original)
  • Use "## Step X:" headers for each step analysis
  • End each step with "This step is correct." or "This step is incorrect."
  • Conclude with "Verification:  Is the answer correct (Yes/No)?  X"
  • Do NOT add any additional sections, reflections, or commentary beyond this format

---

Problem:  __QUESTION_PLACEHOLDER__

Solution Path:  __SOLUTION_PLACEHOLDER__

Original Verification:  __ORIGINAL_VERIFICATION_PLACEHOLDER__

Critique of the Verification:  __CRITIQUE_PLACEHOLDER__

Generate the merged verification in the exact format shown above:

