# OpenReview forum: "SPARK: Stepwise Process-Aware Rewards for Reference-Free Reinforcement Learning"
_ICLR.cc/2026/Conference — Submitted to ICLR 2026_

### Official Review · Reviewer_QmP1 · 2025-10-26

**Soundness:** 3
**Presentation:** 3
**Contribution:** 3
**Rating:** 6
**Confidence:** 4

**Summary:**

This paper introduces SPARK, a three-stage framework to train generative process reward models (PRMs) for reinforcement learning without requiring any ground-truth references or human annotations. In Stage I, it generates a synthetic dataset by using a generator model to create solutions and a more powerful verifier model to evaluate them using inference-time scaling techniques like self-consistency and meta-critique. In Stage II, this synthetic data is used to fine-tune generative PRMs. In Stage III, this reference-free PRM-CoT is used as the reward signal to train a policy model via RL, achieving state-of-the-art results on math reasoning benchmarks, even outperforming baselines trained with ground-truth outcome verification.

**Strengths:**

The primary strength of this work is its novel and effective framework for training process reward models (PRMs) without access to ground-truth references. The reliance on expensive, step-level human annotations or gold solutions is a major bottleneck for scaling process-based feedback, and this paper offers a viable, reference-free alternative. The core idea of using inference-time scaling methods (like self-consistency and meta-critique) to generate high-quality synthetic verification data is a solid contribution.

The empirical results for the PRM itself are strong. The paper shows that a PRM trained on this synthetically generated data (specifically using step-level consistency) achieves a 67.5 F1 on ProcessBench. This result is impressive not only because it outperforms strong LLM critics like GPT-4o, but also because it surpasses a baseline PRM trained with access to ground-truth outcomes, which suggests that aggregating multiple noisy, reference-free process verifications has the potential to provide a richer training signal than verifying against a single ground-truth final answer.

The paper also demonstrates the downstream utility of this reference-free PRM in a practical RL setting and also provides a valuable analysis of reward hacking patterns, and introduces format constraints to mitigate them.

**Weaknesses:**

There is a mismatch between the paper's core motivation and its experimental validation. The method is motivated as a solution for domains where ground truth is "unavailable," "subjective," or "lacks clear verification criteria," such as creative writing or complex planning. However, all experiments are conducted exclusively on mathematical reasoning, a domain defined by objective, verifiable ground truth.

The computational cost of the Stage 1 synthetic data generation pipeline appears to be enormous and is not analyzed. To generate the "Step Consistency" dataset, the framework must run $N=16$ independent verifications for each of the $M=8$ solutions generated for each problem. This implies over 100 verifier passes per problem just to create a single training data point. This massive offline inference cost is not compared against the cost of alternative methods.

**Questions:**

The paper opts for a generative PRM (Gen-PRM) over a discriminative one (Disc-PRM). However, the paper dedicate substantial analysis to reward hacking issues (e.g., solution appending, step inflation) that are unique to this generative approach . Given that Gen-PRMs introduce these complex new failure modes, what is their fundamental advantage over simpler discriminative PRMs that justifies this trade-off?

**Details Of Ethics Concerns:**

The paper does not appear to raise any ethical concerns. It does not involve human subjects, sensitive data, or potentially harmful applications.

---

> ### Author Response · Authors · 2025-11-21
> **Response**
>
> We thank Reviewer QmP1 for their detailed feedback and for recognizing our **novel and effective framework,** our **strong empirical results** on ProcessBench (outperforming GPT-4o and ground-truth baselines), and our **valuable analysis of reward hacking patterns.**
>
> We address the concerns below.
>
> ---
>
> ## Q: Motivation-Validation Mismatch
>
> We agree with the reviewer's concern and clarify our contribution below:
>
> **Why mathematical reasoning as proof-of-concept:** We chose math for rigorous validation: (1) to evaluate whether inference-time scaling methods can generate high-quality verification data for training PRMs, we needed established benchmarks like ProcessBench, which are unavailable for open-ended domains, (2) to validate our method against rule-based RLVR, we needed domains where RLVR is applicable for fair comparison. Math serves as a controlled testbed to establish proof-of-concept with objective evaluation.
>
> **Extension to open-ended domains:** Our framework could potentially extend to domains with semi-objective verification (e.g., science explanations) where multiple critiques via inference-time scaling can identify quality indicators. However, for truly subjective domains like ethical reasoning, our method's effectiveness remains untested. This is an important limitation which we clearly mention in Section 6.
>
> **Revised contribution framing:** Our primary contributions are: (1) first work demonstrating that inference-time scaling generates high-quality step-level verification training data for process reward models, validated on ProcessBench where we **achieve state-of-the-art performance**, surpassing frontier models like GPT-4o, ground-truth methods, and existing PRMs, (2) systematic comparison of reward models (ORM, PRM, PRM-CoT) showing the effectiveness of process-level CoT feedback in resisting ORM-related reward hacking during RL training, (3) proposing effective methods (e.g., Selective Advantage) to incorporate dense step-level feedback in GRPO. We establish proof-of-concept in structured reasoning, which could extend to other domains where step-by-step verification is feasible but costly.
>
> **Section updates:** We will revise Section 1 to focus our motivation on "structured domains where step-level annotations are expensive to obtain (e.g., mathematical proofs, scientific explanations, technical problem-solving)" rather than "subjective domains without ground truth," and clarify in Section 6 that extension to truly subjective domains remains important future work.
>
> ---
>
> ## Q: Computational Cost Analysis
>
> We provide a detailed breakdown of computational costs across all stages:
>
> **Stage I: PRM Training Data Preparation**
>
> | Method | H200 GPU Hours | Token Count | Total Generations → Final Dataset |
> |--------|-----------|------------|--------------|
> | Solution Generation | ~8 | ~35M | 64K solutions |
> | Single Verification | ~18 | ~74M | 64K verifications → 63K examples |
> | Self-Consistency | ~288 | ~1.18B | 1.024M verifications → 63K examples |
> | Meta-Critique | ~54 | ~221M | 192K passes → 63K examples |
>
> **Note:** We use open-source models hosted locally (no API cost), unlike previous methods using closed-source APIs [1,2]. While RLVR avoids reward model training, it requires expensive human annotations, making our automated approach cost-effective at scale.
>
> **Stage II: Reward Model Training**
>
> Training ORM, PRM, and PRM-CoT requires 8, 14, and 26 GPU-hours respectively for supervised fine-tuning.
>
> **Stage III: RL Training**
>
> RL training GPU-hours: ORM (211), PRM (371), PRM-CoT (447), RLVR (150). RLVR has lower cost as it requires no separate reward model, but is only applicable to verifiable domains like math and code.
>
> We will include this complete cost analysis in Section 4.3.
>
> ---
>
> ## Q: Generative PRM vs Discriminative PRM Trade-off
>
> Prior work [1] demonstrates that generative verifiers trained with next-token prediction significantly outperform discriminative verifiers on reasoning tasks, and we observe the same advantage: our generative PRM achieves **67.5 F1 on ProcessBench, surpassing existing discriminative PRMs** (Table 2, Page 17) as well as GPT-4o (61.9 F1). Beyond superior verification accuracy, generative PRMs enable chain-of-thought reasoning that provides explicit step-by-step verification rationales, making them more resistant to reward hacking. While generative PRMs introduce new failure modes like solution appending and step inflation, we systematically address these through format constraints and process-aware reward designs (Section 4.4).
>
> ---
>
> We hope these responses address the reviewer's concerns and would appreciate consideration for revising the score if our clarifications are satisfactory. Thank you for the detailed and constructive review.
>
> **References**
>
> [1] Zhang et al., Generative Verifiers: Reward Modeling as Next-Token Prediction. ICLR 2024
>
> [2] Wang et al., Critique Fine-Tuning: Learning to Critique is More Effective than Imitation. COLM 2024

---

> > ### Author Response · Authors · 2025-11-26
> >
> > Dear Reviewer QmP1,
> >
> > We sincerely hope our responses addressing your concerns about motivation-validation mismatch, computational costs, and the Gen-PRM design choice have been helpful. We greatly value your feedback and would be grateful for any thoughts on our clarifications.
> >
> > Best regards,
> >
> > Authors of Submission 14538

---

### Official Review · Reviewer_9QxJ · 2025-11-01

**Soundness:** 3
**Presentation:** 3
**Contribution:** 2
**Rating:** 4
**Confidence:** 3

**Summary:**

This paper introduces SPARK, a framework to train process reward models (PRMs) for reinforcement learning (RL) entirely without ground-truth references. The method uses a "generator-verifier" system with inference-time scaling (like self-consistency and meta-critique) to create high-quality synthetic step-level verification data. This data is then used to train a generative PRM (PRM-CoT), which subsequently provides reward signals for RL training. SPARK enables reference-free training that outperforms ground-truth methods. On the ProcessBench evaluation, the SPARK-trained PRM achieved 67.5 F1, surpassing the reference-guided (ground-truth) model's 66.4 F1. In RL experiments, SPARK's PRM-CoT led to 47.4% average accuracy, exceeding the ground-truth-based RLVR's 43.9%.

**Strengths:**

1. The design of SPARK is intuitive to me.

1. Instead of relying on a static, expensive ground-truth dataset, SPARK uses a dynamic generator-verifier framework. It leverages inference-time scaling techniques (like self-consistency and meta-critique) to aggregate multiple verification attempts, effectively bootstrapping a high-quality, step-level training dataset from the model's own reasoning capabilities.

1. When used in RL training, SPARK's generative PRM enables the policy model to achieve 47.4% average accuracy on math benchmarks. This result exceeds the performance of the ground-truth-based method, RLVR, which achieved 43.9%.

1. This paper also systematically analyzes reward hacking patterns in process reward-based RL.

**Weaknesses:**

1. The method is motivated by the need to apply RL to subjective domains without ground truth (e.g., creative writing, ethical reasoning). However, all experiments are conducted exclusively in mathematical reasoning, a domain where objective ground truth does exist. This creates a mismatch between the problem the method claims to solve and the domain in which it is actually validated.

1. The paper provides a systematic analysis of reward hacking patterns. However, the identified patterns (e.g., solution appending, step inflation, and step reduction) and their solutions are specific to the highly structured format of mathematical problem-solving, which also deviates from the motivation of applying RL to subjective domains without ground truth. The experimental design does not demonstrate whether these findings or solutions are transferable to the unstructured, open-ended tasks that are the method's ultimate target.

1. The contribution of this paper is marginal. To my understanding, SPARK just combines self-consistency and meta-critique for auto annotation. Besides, there is no quality evaluation to show how accurate the SPARK-generated annotations are.

**Questions:**

1. As mentioned in Weaknesses, how can SPARK be reliably generalized to the very domains the authors use for motivation (like creative writing or ethical reasoning), where an objective verifier doesn't exist and the verifier's critique is just as subjective and unverifiable as the generator's output?

---

> ### Author Response · Authors · 2025-11-21
> **Response Part 1**
>
> We thank Reviewer 9QxJ for their constructive feedback. We appreciate that the reviewer finds SPARK's design **intuitive**, recognizes our **dynamic generator-verifier framework** that effectively bootstraps high-quality training data from the model's own reasoning capabilities, acknowledges our **strong empirical results** exceeding ground-truth-based methods, and values our **systematic analysis of reward hacking patterns**.
>
> We address the reviewer's concerns about the motivation-validation mismatch, quality evaluation of SPARK-generated annotations, and generalization to subjective domains below.
>
> ---
>
> ## Q: Quality Evaluation of SPARK-Generated Annotations
>
> We respectfully direct the reviewer to Section 3.2 (Page 4), where we evaluate PRM quality on ProcessBench, which directly measures the quality of our training data. Our SPARK-trained PRM (14B parameters) **achieves 67.5 F1, outperforming GPT-4o by 5.6 points and exceeding reference-guided training (66.4 F1), and surpassing existing PRMs (Table 2, Page 17)**, demonstrating that our synthetic annotations match or surpass ground-truth-based methods. We also conducted manual verification of annotations to confirm quality, but chose ProcessBench as a scalable evaluation approach.
>
> ---
>
> ## Q: Motivation-Validation Mismatch
>
> We agree with the reviewer's concern and clarify our contribution below:
>
> **Why mathematical reasoning as proof-of-concept:** We chose math for rigorous validation: (1) to evaluate whether inference-time scaling methods can generate high-quality verification data for training PRMs, we needed established benchmarks like ProcessBench, which are unavailable for open-ended domains, (2) to validate our method against rule-based RLVR, we needed domains where RLVR is applicable for fair comparison. Math serves as a controlled testbed to establish proof-of-concept with objective evaluation.
>
> **Extension to open-ended domains:** Our framework could potentially extend to domains with semi-objective verification (e.g., science explanations) where multiple critiques via inference-time scaling can identify quality indicators. However, for truly subjective domains like ethical reasoning, our method's effectiveness remains untested. This is an important limitation which we clearly mention in Section 6.
>
> **Revised contribution framing:** Our primary contributions are: (1) first work demonstrating that inference-time scaling generates high-quality step-level verification training data for process reward models, validated on ProcessBench where we **achieve state-of-the-art performance**, surpassing frontier models like GPT-4o, ground-truth methods, and existing PRMs, (2) systematic comparison of reward models (ORM, PRM, PRM-CoT) showing the effectiveness of process-level CoT feedback in resisting ORM-related reward hacking during RL training, (3) proposing effective methods (e.g., Selective Advantage) to incorporate dense step-level feedback in GRPO. We establish proof-of-concept in structured reasoning, which could extend to other domains where step-by-step verification is feasible but costly.
>
> **Section updates:** We will revise Section 1 to focus our motivation on "structured domains where step-level annotations are expensive to obtain (e.g., mathematical proofs, scientific explanations, technical problem-solving)" rather than "subjective domains without ground truth," and clarify in Section 6 that extension to truly subjective domains remains important future work.
>
> ---
>
> ## Q: Transferability of Reward Hacking Solutions
>
> We respectfully disagree that our solutions are specific only to mathematical reasoning. **Format constraints** (single answer tag, no post-answer content) can impose structure in any domain to prevent exploitation. **Process-aware rewards** with step-level explanations resist hacking better than outcome-only rewards across domains. **Selective advantage** is a general RL technique applicable beyond math. While specific exploitation patterns may vary by domain, the core principle of using process-level verification with explicit rationales provides a general framework for mitigating reward hacking wherever step-by-step verification is feasible.
>
> ---
>
> ## Q: Generalization to Domains with Subjective Verifiers
>
> We acknowledge this important limitation. SPARK relies on the verifier providing reasonable step-level judgments. In truly subjective domains (e.g., ethical reasoning), aggregating subjective judgments may not converge to meaningful consensus. However, many domains have semi-objective quality indicators even without ground truth (e.g., logical consistency, factual accuracy, scientific reasoning), where our approach of aggregating multiple independent critiques can identify common issues. Our work establishes proof-of-concept for structured reasoning which can be extended to semi-verifiable domains. As stated earlier, we will revise our motivation to focus on semi-objective domains rather than truly subjective ones.

---

> ### Author Response · Authors · 2025-11-21
> **Response Part 2**
>
> ## Q: Choice of Self-Consistency and Meta-Critique as Representative Methods
>
> We respectfully disagree that our contribution is marginal because we use self-consistency and meta-critique. Our work establishes a novel framework: **using inference-time scaling to generate high-quality step-level training data for process reward models**. We selected self-consistency (parallel scaling) and meta-critique (sequential scaling) as representative methods from two major categories of inference-time scaling to demonstrate this principle. Future work can explore additional inference-time methods using our methodology.
>
> ---
>
> We hope our responses fully address the reviewer's concerns about motivation-validation mismatch, contribution clarity, and annotation quality. Our work establishes a foundation that enables future research in cost-effective PRM training for domains where step-by-step verification is feasible but costly. If the reviewer finds these clarifications and our repositioned scope satisfactory, we would be grateful for their consideration in revisiting the score. We appreciate the thorough and constructive feedback.

---

> > ### Author Response · Authors · 2025-11-26
> >
> > Dear Reviewer 9QxJ,
> >
> > We sincerely hope our responses addressing your concerns about motivation-validation mismatch, contribution clarity, annotation quality, and transferability have been helpful. We greatly value your feedback and would be grateful for any thoughts on our clarifications.
> >
> > Best regards,
> >
> > Authors of Submission 14538

---

### Official Review · Reviewer_YhFA · 2025-11-03

**Soundness:** 1
**Presentation:** 3
**Contribution:** 2
**Rating:** 2
**Confidence:** 4

**Summary:**

This paper introduces a new method for generating a verification dataset for training PRMs based on scaling a step-wise verification process parallely and sequentially. This synthetic dataset, which comprises step-level judgment with rationales, is used to train these PRMs that later on serves as reward models for RL training. The work presents evaluation on two setups: first, on ProcessBench, claiming that the proposed method for PRM training surpasses GPT-4o and reference-guided verification; and second, on RL training, claiming a performance that matches or exceeds ground truth performance.

**Strengths:**

- The method is conceptually simple and the paper is easy to follow.

- The comparison against GPT-4o and Reference-Guided verification in ProcessBench suggests that the employed methodology is promising, since it does not rely on ground-truth nor on a frontier model.

**Weaknesses:**

- The main concern is the lack of evidence to assess statistical significance in the results. The paper does not mention how many experimental seeds were used (I assume it is a single one), and no results in the paper brings confidence intervals. A well known fact supported by prior literature is that RL training is extremely stochastic [1, 2], which is also observed in math reasoning benchmarking [3], so it is unclear whether the reported takeaways are meaningful or just observation noise. This is particularly necessary to the RL training results (Figs 4-7 at least) but also necessary for the PRM training, as different seeds may lead to different verification performances.

- As the paper states in Section 4.4, one of the limitations from the developed PRMs is that they are still vulnerable to reward hacking, which is one of the main reasons why RLVR is still the standard choice over PRMs. The reward hacking issue is very much present in the “Step-Augmented Process Rewards”, which is the method that strongly relied on step-level rewards.

- The evaluation methodology in Figure 5 is flawed. The Figure highlights a 7.6% difference between PRM-CoT and the other methods, but ignores the fact that prior checkpoints present considerably better performance for ORM. Besides the issue of reporting a single seed, the work also does not provide a methodology on checkpoint selection, and the evaluation on t = 300 is arbitrary.

- The paper does not bring a comparison of the computational cost involved in (1) generating the training dataset; and, more importantly, (2) the cost involved in performing inference in the trained PRMs. From the paper, it is unclear if the proposed PRMs use variable test-time computation, and it would be extremely important to ensure fairness in computation during inference.

- There are other methods to train RL without verifiable rewards, e.g., [4]. It would be nice to compare against them.

Overall, while I see the method as a simple yet interesting direction for generating PRM training datasets, the experimental methodology of the paper is currently flawed which makes the provided evidence weak/questionable. I also believe the main claims of “enabling RL to scale beyond verifiable domains” is somewhat too strong and not supported, especially given that PRMs in general (including the ones in this work) are generally vulnerable to reward hacking, and the proposed method does not address this problem.

**Questions:**

- During verifier inference, are the inference scaling methods also used? Or are they used solely during dataset generation?

- The paper mentions that the datasets contain 63k examples after filtering. Which filtering is that?


References


[1[ Henderson et. al. Deep Reinforcement Learning that Matters. AAAI, 2018.

[2] Agarwal et. al. Deep Reinforcement Learning at the Edge of the Statistical Precipice. NeurIPS, 2021.

[3] Hochlehnert et. al. A Sober Look at Progress in Language Model Reasoning: Pitfalls and Paths to Reproducibility. COLM, 2025.

[4] Zhao et. al. Learning to Reason without External Rewards, 2025.

---

> ### Author Response · Authors · 2025-11-21
> **Response Part 1**
>
> We thank Reviewer YhFA for their thoughtful review. We appreciate that the reviewer finds our method **conceptually simple and easy to follow** and recognizes our ProcessBench comparison as **promising**. We emphasize that our core contribution is synthetic step-level verification data generation via inference-time scaling—**achieves state-of-the-art 67.5 F1 on ProcessBench**, surpassing GPT-4o (61.9), ground-truth baselines (66.4), and existing PRMs (Table 2, Page 17).
>
> We address the reviewer's concerns below.
>
> ---
>
> ## Q: Statistical Significance with Multiple Seeds
>
> We appreciate the reviewer's concern about statistical significance and address this through two clarifications:
>
> **Evaluation metrics clarification:** In our original submission, "Average Test Accuracy" in Figures 4-6 refers to the mean of pass@16 evaluations, averaging 16 independent samples per problem to account for evaluation randomness. This is standard practice in the community for challenging test sets such as AIME. We will clarify this in the captions of Figures 4-6 and in Section 4.2.
>
> **Multiple training seeds:** The reviewer is correct that our original results used single training runs. While multiple training runs with different seeds are not standard practice in the community (which typically reports results from a single training run), we have re-run all RL experiments (Figures 4-6) with 4 different random seeds to address this concern. **Updated Figures 4-6 now show mean performance with shaded confidence intervals across all seeds, showing the same trends as the original figures and demonstrating the robustness of our results.**
>
> We hope these additional experiments and clarifications fully address the reviewer's major concern about statistical rigor.
>
> ---
>
> ## Q: Computational Cost Analysis
>
> We provide a detailed breakdown of computational costs across all stages:
>
> **Stage I: PRM Training Data Preparation**
>
> | Method | H200 GPU Hours | Token Count | Total Generations → Final Dataset |
> |--------|-----------|------------|--------------|
> | Solution Generation | ~8 | ~35M | 64K solutions |
> | Single Verification | ~18 | ~74M | 64K verifications → 63K examples |
> | Self-Consistency | ~288 | ~1.18B | 1.024M verifications → 63K examples |
> | Meta-Critique | ~54 | ~221M | 192K passes → 63K examples |
>
> **Note:** We use open-source models hosted locally (no API cost), unlike previous methods using closed-source APIs [1,2]. While RLVR avoids reward model training, it requires expensive human annotations, making our automated approach cost-effective at scale.
>
> **Stage II: Reward Model Training**
>
> Training ORM, PRM, and PRM-CoT requires 8, 14, and 26 GPU-hours respectively for supervised fine-tuning.
>
> **Stage III: RL Training**
>
> RL training GPU-hours: ORM (211), PRM (371), PRM-CoT (447), RLVR (150). RLVR has lower cost as it requires no separate reward model, but is only applicable to verifiable domains like math and code.
>
> We will include this complete cost analysis in Section 4.3.
>
> ---
>
> ## Q: Checkpoint Selection Methodology
>
> We appreciate the reviewer raising this important question and address this through multiple analyses:
>
> **(1) Equal compute comparison:** We evaluate at t=300 across all experiments to ensure consistent comparison under equal training compute budget.
>
> **(2) Best and average checkpoint comparison:** PRM-CoT's best checkpoint (t=240) achieves **4.1% higher accuracy than ORM's best checkpoint (t=80)** with statistical significance (p<0.05, averaged over 16 evaluations). Comparing average performance across checkpoints (every 20 steps) shows PRM-CoT maintains 4.5% advantage over ORM.
>
> **(3) Extended training analysis:** We extended training beyond 300 steps and find PRM-CoT continues improving while ORM degrades. Transcript analysis reveals the policy exploits ORM's outcome-only signals, whereas PRM-CoT's step-level explanations of correctness resist exploitation.
>
> We will include the best checkpoint analysis in Section 4.3 and extended training observations in Section 4.4.
>
> ---
>
> ## Q: Comparison with Alternative Reference-Free RL Methods
>
> In our original submission, we did not include INTUITOR [3] as preliminary experiments showed it exhibits training collapse. However, we now include this comparison (Appendix Fig 7, Page 19), confirming that INTUITOR's self-certainty based reward causes catastrophic performance collapse as the model learns to generate confidently wrong answers with high certainty.
>
> ---
>
> ## Q: Reward Hacking Vulnerability
>
> As the reviewer noted, Step-Augmented Process Rewards exhibits performance degradation due to reward hacking, which we clearly describe in Section 4.4. However, our comparison shows varying vulnerability across PRM-CoT variants: Process-Aware reward maintains consistent improvement without hacking, while Selective Advantage and Global Step Reward also remain stable (Fig 6). Our Process-Aware approach with step-level explanations effectively mitigates the reward hacking issue.

---

> ### Author Response · Authors · 2025-11-21
> **Response Part 2**
>
> ## Q: Dataset Filtering Process
>
> We filter to 63K examples by removing incomplete verifications and ensuring equal dataset sizes across all methods for fair comparison. For step-level majority voting, we select verifications matching the consensus judgment pattern across all steps. In cases where no exact match exists, we exclude the example from all methods.
>
> We will include these details in Section 3.1.
>
> ---
>
> ## Q: Inference Scaling Usage
>
> Inference-time scaling methods (self-consistency, meta-critique) are used **only during Stage I dataset generation**. Once trained, our PRMs perform **single-pass inference** during both ProcessBench evaluation and RL training, generating one verification per solution.
>
> ---
>
> ## Q: Claim Regarding RL Beyond Verifiable Domain
>
> We agree with the reviewer's concern about the motivation-validation mismatch and clarify our contribution below:
>
> **Why mathematical reasoning as proof-of-concept:** We chose math for rigorous validation: (1) to evaluate whether inference-time scaling methods can generate high-quality verification data for training PRMs, we needed established benchmarks like ProcessBench, which are unavailable for open-ended domains, (2) to validate our method against rule-based RLVR, we needed domains where RLVR is applicable for fair comparison. Math serves as a controlled testbed to establish proof-of-concept with objective evaluation.
>
> **Extension to open-ended domains:** Our framework could potentially extend to domains with semi-objective verification (e.g., science explanations) where multiple critiques via inference-time scaling can identify quality indicators. However, for truly subjective domains like ethical reasoning, our method's effectiveness remains untested. This is an important limitation which we clearly mention in Section 6.
>
> **Revised contribution framing:** Our primary contributions are: (1) first work demonstrating that inference-time scaling generates high-quality step-level verification training data for process reward models, validated on ProcessBench where we **achieve state-of-the-art performance**, surpassing frontier models like GPT-4o, ground-truth methods, and existing PRMs, (2) systematic comparison of reward models (ORM, PRM, PRM-CoT) showing the effectiveness of process-level CoT feedback in resisting ORM-related reward hacking during RL training, (3) proposing effective methods (e.g., Selective Advantage) to incorporate dense step-level feedback in GRPO. We establish proof-of-concept in structured reasoning, which could extend to other domains where step-by-step verification is feasible but costly.
>
> **Section updates:** We will revise Section 1 to focus our motivation on "structured domains where step-level annotations are expensive to obtain (e.g., mathematical proofs, scientific explanations, technical problem-solving)" rather than "subjective domains without ground truth," and clarify in Section 6 that extension to truly subjective domains remains important future work.
>
> ---
>
> We hope our responses, including new multi-seed experiments with confidence intervals, comprehensive computational cost analysis, comparison with alternative reference-free RL methods, and clarifications on checkpoint selection and evaluation metrics, fully address the reviewer's concerns about statistical rigor and experimental methodology. If the reviewer finds these substantial improvements satisfactory, we would be grateful for their consideration in revisiting the score. Thank you for the thoughtful and constructive evaluation.
>
> **References**
>
> 1. Zhang et al., *Generative Verifiers: Reward Modeling as Next-Token Prediction*, ICLR 2024
>
> 2. Wang et al., *Critique Fine-Tuning: Learning to Critique is More Effective than Imitation*, COLM 2024
>
> 3. Zhao et al., *Learning to Reason without External Rewards*, arXiv 2024

---

> > ### Author Response · Authors · 2025-11-26
> >
> > Dear Reviewer YhFA,
> >
> > We sincerely hope our detailed responses addressing your concerns about statistical significance, computational costs, checkpoint selection, and alternative baselines have been helpful. We greatly value your feedback and would be grateful for any thoughts on our clarifications.
> >
> > Best regards,
> >
> > Authors of Submission 14538

---

### Meta-Review · Area_Chair_6jZH · 2025-12-23

**Summary:**

SPARK is a three-stage framework for reference-free reinforcement learning that trains generative process reward models without requiring ground-truth annotations. By leveraging inference-time scaling techniques like self-consistency and meta-critique, the system generates high-quality synthetic step-level verification data. These models then serve as reward signals during reinforcement learning. Experimental results on mathematical reasoning tasks demonstrate that this approach can outperform traditional ground-truth-based methods.

**Strengths:**
1. The method is conceptually simple and effective, using inference-time scaling to create high-quality synthetic verification data without ground truth. It offers a viable alternative to expensive step-level annotations, addressing a major scaling bottleneck for process-based feedback.
2. The empirical results are strong, with the trained PRM surpassing GPT-4o and reference-guided baselines on the ProcessBench evaluation, and the downstream reinforcement learning performance on math benchmarks exceeding established ground-truth-based methods such as RLVR.

**Weaknesses:**
1. There is a significant mismatch between the core motivation of applying reinforcement learning to subjective domains and the experimental validation conducted solely on objective mathematical tasks. It remains unproven whether the identified reward hacking patterns or the proposed format constraints can transfer to unstructured, open-ended tasks.
2. The experimental methodology lacks statistical significance as the paper relies on single-seed runs without providing confidence intervals or clear checkpoint selection criteria. Therefore, it is difficult to determine if the reported improvements over baselines are meaningful or merely observation noise.
3. The paper fails to provide a thorough analysis of the substantial computational costs associated with the offline synthetic data generation pipeline, hindering fairness in comparison.

**Reviewer Concerns:**

During the rebuttal, the authors addressed several key technical concerns by providing statistical significance through results from multiple experimental seeds and detailing a clear checkpoint selection methodology. They also provided a comprehensive breakdown of computational costs across all stages and included comparisons with alternative reference-free RL methods to better contextualize their performance. To address the issue of reward hacking, the authors argued that their process-aware approach, which incorporates step-level explanations, serves as an effective mitigation strategy. They also attempted to address the broader concern regarding the application of reinforcement learning beyond verifiable domains.

**Reviewer Scores:**

The mismatch between the paper’s primary motivation and its empirical validation remains a significant concern raised by all reviewers. While the authors justify using mathematical reasoning as a proof-of-concept and argue that the framework could potentially extend to domains with semi-objective verification, a purely conceptual explanation may not be sufficient to resolve reviewer doubts without supplementary experiments in other fields. Similarly, the transferability of reward hacking solutions and the system's generalization to domains with subjective verifiers remain critical issues. It may be difficult to fully convince the reviewers of the framework’s broader utility through argumentation alone, as these aspects require empirical evidence to prove they can function in truly unverifiable or unstructured contexts. Considering the above situation, the reviewers may increase the score, but the extent of the increase is likely to be marginal.

---

### Decision · Program_Chairs · 2026-01-26

Reject